## Comment

 

**Subject Category:**
Biology (whole organism)

evolution/behaviour

**Author for correspondence:**
Peter Nonacs
e-mail: pnonacs@biology.ucla.edu

# Hamilton's rule is essential but insufficient for understanding monogamy's role in social evolution

## Peter Nonacs

Department of Ecology and Evolutionary Biology, University of California, Los Angeles, CA 90095, USA

iD PN, 0000-0003-3010-0777

In 2010, Nowak *et al.* ([1]: hereafter cited as NTW) fired a broad salvo at the concept of inclusive fitness, criticizing the underlying mathematics and proclaiming that the conceptual idea itself had produced 'meagre' benefits towards understanding the evolution of cooperative living. The response to their paper was swift and overwhelmingly negative, including one rebuttal of almost every point that bore the signatures of 134 co-authors [2]. Criticisms of and responding defences for inclusive fitness methodologies and conclusions have not abated and arguably have even intensified. Neither side appears much convinced by the arguments of the other.

The debate has also at times narrowed from broad criticism like NTW to a series of more specific questions within social evolution. No single topic better exemplifies the on-going controversy than the question of what role monogamy plays in the evolution of reproductive division of labour within cooperating family associations. This ranges from the initial decisions of offspring to either disperse or remain as subordinate helpers for their parents in facultatively social species [3–8], to the appearance of a morphologically sterile worker caste in obligately social species [9–12].

That monogamy might be important flows from a direct application of Hamilton's rule, where a trait is selectively favoured whenever the inequality of $br - c > 0$ is met. In this equation, the cost ($c$) an actor suffers in terms of lost direct offspring (related by 0.5) must be offset by the number of offspring ($b$) that are produced due to the actor's actions, prorated by the genetic relatedness ($r$) of those offspring to the actor. Thus, the summed effect on fitness for expressing a given trait is inclusive of the effect it has on both the bearer of the trait and the consequences of the bearer's interactions with genetic relatives. Clearly, this inequality is more easily satisfied when involving the production of full siblings ($r = 0.5$ under

monogamy) than with a collection of full and half-sibs ($r < 0.5$) as from multiple mating or cooperative breeding by multiple individuals. Consistent with the equation's predictions, phylogenetically controlled comparative analyses implicate monogamy, or at least reduced promiscuity, as the most probable ancestral states across social Hymenoptera, cooperatively breeding birds, and social mammals [3,4,8]. Also, some studies that attempted to measure the inclusive fitness of helpers found that Hamilton's condition for advantageous helping is met [13,14]. Indeed, the apparent necessity that the evolution of both facultative and obligate sterility must pass through a 'monogamy window' was presented as a clear refutation of NTW [15].

Subsequently, however, this seemingly very straightforward prediction from Hamilton's rule did not always similarly arise when modelled in population contexts [6,7,10,11,16]. Furthermore, the above comparative analyses, while supporting high relatedness as the most probable ancestral state, also suggest numerous later evolutionary transitions to multiple mating and reduced within-group genetic relatedness—without any recorded increase in group disharmony. Thus, it appears that for unexplained reasons Hamilton's rule is a unidirectional phenomenon that only applies to increasing cooperation and not to its diminishment.

New empirical studies also cast doubt on the degree to which helping behaviour actually benefits the helper's overall fitness. For example, in many Hymenopteran species helping behaviour by daughters is facultative in that all are fully reproductively capable. Therefore, those that remain with their mother have apparently chosen to voluntarily become non-reproductive workers. Careful measures of nest reproductive success across multiple species, however, found fitness gains for helping relatives to be significantly less than expected gains from dispersing and reproducing one's own offspring [17–21]. Somehow very cooperative, yet also apparently very maladaptive behaviour continues to persist in these populations. At a minimum, this calls into question the idea of freely chosen 'helper' roles [22–24].

We can start to make sense of this dissonance in theory and experimental result by first examining why the models disagree on the effects of monogamy. In considering a daughter's choice between being a non-reproductive helper for her mother or dispersing, Nonacs [6] and Fromhage and Kokko [5] created seemingly identical models. However, they came to almost diametrically opposite conclusions. Nonacs finds that under most conditions the helper trait will spread faster when females mate with multiple males. By contrast, Fromhage and Kokko find that both monogamy and haplodiploidy facilitate the spread of helping. It turns out that these divergent outcomes are driven by one difference in their assumptions. Both models consider the possibility of the mother dying during offspring rearing. Nonacs assumes that a surviving helper promotes to the mother's role and takes over reproduction; Fromhage and Kokko assume that the group perishes with the mother. Removing promotion from the Nonacs model, and thus any direct fitness for helping, produces results identical to Fromhage and Kokko, and vice versa [10]. Therefore, the effect of monogamy crucially depends on whether or not a species' life history and ecology creates opportunities for helpers to also gain a selfish benefit in producing their own offspring.

In considering the evolution from facultative to obligate sterility in insect worker castes, Nonacs [10] emphasizes the importance of which actors are most probable to impose sterility. If the workers themselves are acting in a self-sacrificial manner, or older siblings impose sterility on younger ones and thereby increase the queen's reproduction, then monogamy is the more favourable preadaptation. Conversely, matedness level has no effect if maternal manipulation imposes sterility. Finally, multiple mating is the more favourable preadaptation if worker reproduction is already restricted to producing only sons and its suppression leads to little or no gain in reproduction for the queen. Olejarz *et al*. [11] also considered a scenario where all workers are unmated and capable of producing only sons, and their production might compete with the reproduction of sons by the queen. Under this scenario, there are parameter combinations where voluntary worker-sterility alleles invade only with monogamy, only with multiple mating, or under either condition. Davies & Gardner [12] expanded this analysis by allowing sterility alleles to be intermediate in their effect, the evolution of sterility beyond invasion criteria, and with a broader range of worker sterility scenarios. Under these differing conditions, there were no parameter combinations such that only multiple mating favoured sterility. However, for an even broader set of parameter values, matedness levels had no apparent significant effect (e.g. figs. 1b and 2b in [12]).

In summary, the effects of monogamy (and therefore, relatedness) on the evolution of reproductive division of labour in cooperatively breeding groups have been examined by a variety of methodologies and assumptions. An easily drawn overall conclusion is that monogamy can enhance, retard or have no effect on the willingness of individuals to sacrifice their own reproduction. This

variation in predicted outcomes often critically depends on assumptions or ecological factors that have nothing to do with how often parents mate. Combining all the results gives a fuller, but undoubtedly not a complete picture of the evolutionary effect of monogamy.

Unfortunately, the relationships across papers have been rather more antagonistic than synergistic. The follow-on papers from NTW tend to use their results as a cudgel to bash the entire concept of inclusive fitness and the mathematics associated with it [11,25–30]. If Hamilton's rule fails or is irrelevant under specific conditions, that result is extrapolated into a general denunciation of inclusive fitness. However, none of these studies show that general predictions derived from the basic inequality of the Hamilton equation are universally wrong. For example, sacrificing one's own reproduction to help a full sibling reproduce will reap higher fitness returns than helping a half-sib reproduce. Also, if inclusive fitness represents a flawed and bankrupt methodology, then it is odd how often its general predictions receive strong empirical support [31,32].

Equally unfortunate is that responses to NTW and its successors vary from categorical rejection [2,33–35], to adopting a prosecutorial style that mostly emphasizes revealing the limited special case nature of the critical models [12,36–38], to avoiding engaging with inconvenient results by raising disingenuous objections about assumptions and methods (as [7] and [16] claim about [6]). None of these defenders of inclusive fitness, however, has cited any actual errors in the mathematics of the studies they criticize. In short, a special case scenario may be inappropriately generalized, but as exceptions continue to accumulate this does weaken the case for immediate acceptance of predictions derived directly from Hamilton's inequality.

The way forward is what Okasha [39] presciently recommended immediately in the wake of NTW. I would summarize it as the realization that everyone is right – to the extent that nature reflects the underlying assumptions of their models. Alternative models serve alternative purposes. Without Hamilton's insight and inclusive fitness model, would there be any discussion about the potential importance of monogamy for social evolution? Without the evolutionary simulation techniques, would there be the mathematical tools to effectively accommodate a variety of ecological constraints and life histories simultaneously with optimal mating behaviour? Amalgamating all the past work into a single complementary division of research labour opens multiple opportunities. Future theoretical and empirical research on the effect of relatedness (i.e. monogamy) and how it may interact in varied and potentially surprising ways with life history, ecology and gene expression could consider:

(1) Power in the degree to which group members can affect each other's roles [40]. For example, if helper roles are imposed on offspring by parents, monogamy or not is irrelevant. Such may be the case in the facultatively social halictid bee, *Megalopta genalis*, where maternal manipulation through feeding regimen is strongly implicated [23,41], and the frequency of social versus non-social nests in a natural population corresponds closely to that predictions of models that presume maternal manipulation [19]. Conversely, if older siblings determine the fecundity of younger ones, then parental mating behaviour may be very influential [10,11].

(2) Genetic diversity versus kin nepotism. High relatedness within a group helps guarantee that benefits of sociality flow to shared genes. However, this comes at a cost of lowered diversity for social heterosis [42]. When simulated, the degree to which groups can benefit from being diverse easily selects for behavioural adaptations that reduce within-group relatedness such as multiple mating and the 'drifting' of non-reproductive helpers between unrelated groups as a form of indirect reciprocity where all colonies gain the benefits of genetic diversity through exchanging a portion of their workforce [43].

(3) Does within-group conflict increase with evolutionary transitions away from monogamy? For example, obligate sterility in the worker caste of social insects is assumed to be a morphologically irreversible state [9]. Sterile workers, however, have the same intact and viable genes for reproduction as do queens. Consider if sterility results from mutations in gene regulatory processes in development, and that such mutations are advantageously selected only with monogamy. Then the loss of monogamy ought to create situations where mutations back to fertility would be adaptive, with the expectation that obligate sterility has not always been evolutionarily irreversible. Furthermore, considerable effort has gone into finding correlations between high relatedness and increased cooperation. Hamilton's rule, however, should work in both directions: creating evident patterns of increased conflict and reduced reproductive sacrifice across the many demonstrated evolutionary transitions from higher to lower relatedness in group structure [3,4].

(4) Direct versus indirect fitness. If subordinate helpers or workers sometimes have a realistic chance of reproducing, this may reduce the importance of sib-sib relatedness. One such example is in polistine

wasps, where the possibility of inheriting the nest and becoming the dominant reproductive predicts cooperation while waiting in a queue [44]. This is likely to be particularly relevant across long-lived, multigenerational groups, such as humans [45].

(5) Within-group competition for resources or breeding opportunities. Theoretically, sometimes the best strategy for closely related kin may be to just get out of each other's way [7]. Ecologically, the expected success from dispersing versus how much a helper can add to group productivity may have a greater impact than variation in within-group relatedness.

(6) Within-group incest avoidance. High relatedness may become problematical for multigenerational cooperative groups due to inbreeding. If this results in sex-biased dispersal across groups, one would not expect similar levels of either reproductive skew or cooperative effort across the sexes [46].

In summary, it is hard to imagine that the diversity of questions about social behaviour could have arisen without the guidance of Hamilton's rule and the concept of inclusive fitness. However, it is just as important to realize that relying solely on inclusive fitness type models will severely limit the degree to which any of the above suggestions can be pursued. As is amply demonstrated, ecology, genetics and group-level benefits all complicate the simplicity of Hamilton's rule. Thus, in the modified words of a former US President, 'Trust in inclusive fitness, but verify'.

Data accessibility. This article has no additional data.

Competing interests. I declare that I have no competing interests.

Funding. No research support funding is involved with this work.

Acknowledgements. I sincerely thank K. M. Kapheim, N. G. Davies and J. J. Boomsma for their helpful comments and suggestions on previous versions. This, however, does not imply agreement with any or all the opinions expressed.

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
