## [Reviewer comments · Royal Society Open Science]

Review History

RSOS-180913.R0 (Original submission)

Review form: Reviewer 1 (Nicholas Davies)

Is the manuscript scientifically sound in its present form?

Yes

Are the interpretations and conclusions justified by the results?

Yes

Is the language acceptable?

Yes

Is it clear how to access all supporting data?

Not Applicable

Do you have any ethical concerns with this paper?

No

Have you any concerns about statistical analyses in this paper?

No

Recommendation?

Accept with minor revision (please list in comments)

Comments to the Author(s)

In this comment, Prof. Nonacs gives his view of the inclusive fitness controversy and related issues, partly through the lens of two recent papers: Olejarz et al (2015, eLife) and Davies and Gardner (2018, Royal Society Open Science).

Olejarz et al (2015) analysed a model of the evolution of voluntary worker non-reproduction in haplodiploid insect societies and found that monogamy sometimes promotes and sometimes inhibits the evolution of worker sterility. Davies and Gardner (2018) analysed the same model and found that the results of Olejarz et al. were due to restrictive assumptions concerning the genetics of worker sterility. When we (Davies and Gardner) relaxed these assumptions, we found that monogamy actually has a clear promoting effect on the evolution of worker sterility.

I have to say that I don't agree entirely with Prof. Nonacs' interpretation of my paper, but I do appreciate that he is trying to propose a way forward here and so I'll mainly restrict myself in this review to commenting on what I see as a few small matters of factual correctness. I don't think any of these problems are fundamental to what is being said in this comment piece more generally, but do need to be addressed.

Lines 82–84: "Davies and Gardner [24] expanded this analysis by allowing sterility alleles to be intermediate in their effect and beneficial to the queen in allowing her to gain more reproduction."

The first part of this sentence is right, but the second part doesn't seem right to me. Olejarz et al. made the assumption that the p_z function (proportion of male eggs produced by the queen) would be an increasing function of z . We adopted the same assumption. Therefore I don't think this can be described as a way in which we extended the analysis of Olejarz et al. To be clear, we did extend their analysis, but we did so by (1) allowing sterility alleles to be intermediate in their effect, (2) considering the long-term evolution of worker sterility rather than the short term invasibility of particular alleles, and (3) looking at a broader range of worker sterility scenarios.

On a related note, further up in this paragraph, it is stated that Olejarz et al. assumed that "the production of [worker-derived] males competes with and possibly reduces the number of males that are queen-produced." This is technically right but I think phrased in a way that could be clearer. Olejarz et al. made the assumption that queen male production increases with worker sterility, so I feel like the word "possibly" isn't quite right here. Overall production depends on both r_z and p_z , so you *could* get a situation where worker sterility causes everyone's production to decrease (decreasing r_z), which *could* reduce the queen's male production overall (depending on p_z), but I think overall the way this is described here might be a little confusing to readers.

Lines 92–94: Davies and Gardner [24] respond to Olejarz et al [12] by declaring that "monogamy always promotes... worker sterility" without qualifying that this requires traits to be expressed additively..."

We spend quite a lot of the paper exploring alternative scenarios in which worker sterility is

dominant or recessive, so I don't know where this is coming from – we absolutely do not require traits to be expressed additively.

Lines 124–126: "The degree to which parents or siblings can impose complete or almost complete sterility on their group mates will determine whether the Olejarz et al. [12] or Davies and Gardner [24] models are more appropriate."

I'm not sure I understand this – we make the same assumptions as Olejarz et al when it comes to who is controlling sterility, so I don't think whether parents or siblings can impose sterility on their group mates has anything to do with whether our analysis is more appropriate than Olejarz et al.'s.

Finally, I wonder if the author would consider toning down the last paragraph just a little. I have always appreciated Prof. Nonacs' point of view, which has come to me through peer review a couple of times, and I appreciate that he makes his points in a measured way and does not seek to bury papers that he disagrees with. I also agree in general with the assessment in this paragraph that diversity of opinion can be fruitful and that excessive dogmatism is unhelpful. But I don't think I or my coauthor can fairly be described as people who consider "proving their opponents wrong" to be the most important thing, and it feels a bit like that is what is being said here. I worked on this paper during my PhD and I would never want to discourage anyone from wanting to study social evolution. In fact, I met with Carl Veller (one of the authors on Olejarz et al, who was a PhD student like me at the time) in person to ask him for feedback on the paper, and because I didn't want the authors – especially the less senior ones – to feel like there was any bad blood between us. So although I'm sure this wasn't Prof. Nonacs' intention, the final paragraph feels a little bit unfair to me.

Best wishes,

Nick Davies

Review form: Reviewer 2 (Jacobus J. Boomsma)

Is the manuscript scientifically sound in its present form?

No

Are the interpretations and conclusions justified by the results?

No

Is the language acceptable?

Yes

Is it clear how to access all supporting data?

Not Applicable

Do you have any ethical concerns with this paper?

No

Have you any concerns about statistical analyses in this paper?

No

Recommendation?

Major revision is needed (please make suggestions in comments)

Comments to the Author(s)

This manuscript evaluates the extent to which monogamy promotes altruistic cooperation. It is not completely clear to me why Peter Nonacs appears to feel that there are scientific reasons for considering possible alternatives to inclusive fitness theory for explaining patterns and processes of social evolution, which he does in practice in spite of his closing sentence. Apart from the reference to Nowak et al (2010) nothing in his own work (which appears in several hypothesis-ideas towards the end of the manuscript) seems to justify that. I therefore believe that much of this essay sees controversy and disagreement where there in fact is none, provided one is clear about terms and definitions and the ways in which they were messed up when sociobiology became influential.

In the four comments below I attempt to clarify this overall assessment. After that I provide specific comments with reference to line numbers:

1. Nowak et al. (2010) has nothing to do with the topic at hand. The reference to the Nowak et al. (2010) paper at the start of this manuscript is odd. That paper ignored the monogamy hypothesis (in spite of two Boomsma reviews of 2007 and 2009) having laid it out in considerable detail and the Science paper by Hughes et al. (2008) having shown it was supported by considerable empirical evidence. The Nowak et al. challenge of inclusive fitness theory was thoroughly refuted by a series of 5 rebuttals of which only one (Boomsma et al., 2011) explicitly addressed monogamy – the total number of authors involved was ca. 150 (not 134 as Nonacs states). The agenda of the Nowak et al. paper failed, because a scientific challenge of a major paradigm needs to achieve two things: 1. It needs to allow easy understanding of all insights already validated under the previous paradigm (in this case the 16 subfields listed with triple ‘yes’ in Table 1 of the Abbot et al. rebuttal that summarized the overwhelming empirical and modelling confirmation of inclusive fitness theory), and 2. Such a challenge needs to add a series of new testable predictions that were not valid questions under the existing paradigm, and that would have sufficient initial support to open up entire new research fields. This is what relativity theory (Einstein) and quantum theory (Bohr) did when they built on and enriched Newton mechanics. Nothing like that was achieved by Nowak et al. (2010). Their statement of ‘meagre’ support is simply wrong. Neither did their paper allow the recapture of any previous understanding or generate a single novel testable prediction. It therefore seems inappropriate to start the introduction of the present manuscript with Nowak et al. (2010). It would also be nice if Nonacs could actually mention more explicitly that the monogamy hypothesis has been very well supported by empirical data, not unexpectedly as it is in fact just an extension of Hamilton’s rule and the inclusive fitness theory behind.

2. Nonacs’ formulation of the central prediction of the monogamy hypothesis is incorrect. Nonacs appears to misrepresent what the monogamy hypothesis actually predicts. He is not the only one, because the Hughes et al. (Science 2008) paper testing the empirical evidence as provided by the then available comparative data is actually not very precise in how it phrased the idea being tested. If Nonacs is interested in evaluating the status of the monogamy hypothesis, he should therefore go back to its exact formulation by Boomsma, both in the first review cited by Hughes et al. (Current Biology 2007) and reread the precise phrasing in the review paper that he cites (Boomsma 2009). That should reveal that the monogamy hypothesis was ONLY meant to explain the evolution of life-time physically differentiated castes, not the evolution of all altruistic cooperation as the title of the present Nonacs manuscript suggests. It was stated explicitly that this concerned the evolution of superorganismality *sensu* Wheeler, which Boomsma coined ‘obligate eusociality’ struggling with the ‘eusociality definition’ that Wilsonian sociobiology had left the field with. It has become increasingly clear (and has been expressed in increasingly explicit

language by Boomsma (2009, 2013), Boomsma, Huszár & Pedersen (2014), and Boomsma & Gawne (2018) that sociobiology terminology is a smokescreen because its 'eusociality' does not discriminate between first helpers at the nest and life-time differentiated castes. The monogamy hypothesis never claimed to explain all aspects of what determined first helpers at the nest. It is about predicting the emergence of obligate altruism, not of facultative (condition-dependent) forms of altruism (facultative eusociality in the Boomsma reviews). Nonacs has made this interpretational mistake in his earlier modelling papers, which he now admits (e.g. nest inheritance), but his present text continues to repeat some of the same misapprehensions. His arguments thus acquire several strawman characteristics and they are already apparent in the title of the manuscript (the monogamy hypothesis is not about cooperation because reproductive altruism \neq cooperation).

3. Assumptions should be explicit and biologically realistic for models to be meaningful. It is of crucial importance for evaluating the assumptions that go into any model meant to describe the evolutionary origin of different types of altruistic phenotypes. The previous Nonacs models have always addressed the origin of helpers at the nest, not the origin of obligatory altruistic castes, i.e. the irreversible transition to Wheeler-superorganismality which implies that ALL colony members belong to a single adult caste phenotype for life. The monogamy hypothesis never intended to challenge the recurrent origins of facultative (condition-dependent) altruism through the simple logic of Hamilton's rule ($rb > c$). If all three variables vary freely there can be situations where lower than full-sib relatedness still allowed some offspring to become helpers at the nest, for example because they were poorly endowed as adult phenotypes (maternal manipulation could play a role here, but note that mothers also need to weigh how they get most genes passed on to grand-offspring by making offspring disperse rather than stay). However, what the life-time monogamy hypothesis states is that it cannot be expected that this will ever lead to the evolution of permanently unmated workers (true workers as termite researchers aptly call them; neuters as Darwin and Wheeler conceptualized them). The monogamy hypothesis is about the evolution of obligate (unconditional) altruism, a distinction that is not made in the Hughes et al (2008) paper but that is explicit in the Boomsma reviews. Most examples that Nonacs highlights are about the origin of first helpers at the nest, and none is about helping behaviour going to fixation so that physically differentiated permanent castes appear. If Nonacs would make that distinction, most of the rationale for his critique would seem to disappear. The same rationale as outlined here also pervaded the recent Davies and Gardner paper, so the Nonacs critique appears to compare apples and oranges. A recent modelling paper by Quinones & Pen (Nature Communications, 2017) also shows clearly that strict monogamy is a necessary condition for the evolution of obligate, not facultative reproductive altruism. The Nonacs (2011, 2014) papers thus appear to have their assumptions wrong for them to be relevant as tests of the monogamy hypotheses for permanent castes – they address the evolution of first helpers.

4. The proposed 'alternative' hypotheses towards the end of the manuscript. Some of these appear to be interesting research agendas, but others seem rather far-fetched and lack more than hand-waving justification. More importantly, however, very few if any of them are directly relevant for testing the monogamy hypothesis in its original meaning.

I hope the following line-by-line comments will clarify these general points.

Line 26: 'eusociality' here lumps societies and superorganisms, which Boomsma & Gawne show are fundamentally different social systems, going back to Darwin, Weismann, Wheeler, Fisher, Williams and others all the way through the 1960s, until sociobiology started to water down and muddle these definitions. The Boomsma reviews (particularly 2013) also make this point, emphasizing that Crespi & Yanega (1995) and Beekman et al (2006) also criticized the 'eusociality continuum' ideas emanating from the watered down definitions by E.O. Wilson in the 1970s.

Line 29: Refs 9+10+12 had their modelling assumptions wrong for the monogamy hypothesis as originally formulated by Boomsma, which ref. 12 partly rectifies and the recent Davies&Gardner paper makes more explicit. You also admit that later on, but without stating the explicit reasons. There is also a semantic issue here. Empirical data can never be 'put to mathematical tests', not even in physics. Mathematical models can be used to derive predictions from axiomatic assumptions that can then be empirically tested, not the other way around. Being precise about this distinction is crucial particularly in evolutionary biology, where the main function of models is to make sense of empirically observed variation in social behaviours and life-histories – anything beyond that is illusionary.

Line 30: The monogamy hypothesis is not about the evolution of cooperation (for mutual benefits) but about the evolution of obligate reproductive altruism. What Hughes et al. (2008) and Cornwallis et al. (2010) tested was whether low parental promiscuity favors facultative reproductive reproductive altruism (with for social insects obligate reproductive altruism as a putative later extension). They never tested the hypothesis in its most stringent version originally proposed.

Line 32-33: When you conceptualize the evolution of permanent true worker castes as an evolutionary point of no return (i.e. a major evolutionary transition), as all Boomsma reviews since 2007 have done, the later transitions to multiple queen mating and polygyny (the latter primarily in ants – see Boomsma, Huszár & Pedersen, *Animal Behaviour* 2014) are not relevant for the argument, because by then true worker castes could no longer go back to facultative (condition-dependent) altruism. Also here 'cooperative behaviour' is an inaccurate description of what the monogamy hypothesis is about (same in Line 35).

Line 35-40: Once more, the monogamy hypothesis is not about these cases. If daughters stay to help even though they could have dispersed, there is no reason to question that they do that because the inequality specified by Hamilton's rule compels them to do so. Because these cases represent facultative altruism, it is expected that measured average direct and indirect fitnesses are in fact equal (representing a frequency dependent ESS equilibrium where both strategies do equally well). This is what the cited studies report and it is all consistent with inclusive fitness theory, even though the papers try to argue otherwise – very little if anything in these decision making processes can be shown to be maladaptive or challenging established inclusive fitness theory (line 41-42). But the main issue is that none of this challenges the monogamy hypothesis for its prediction of major evolutionary transitions to superorganismality as defined by Wheeler (Boomsma & Gawne, 2018).

Line 44: The above implies there is no dissonance between theory and empirical results as implied here, provided one phrases the predictions and assumptions of the Boomsma hypothesis correctly, rather than forcing them into uninformative sociobiology jargon that Boomsma tried to avoid.

Line 50: Again, the argument is not about cooperation but about reproductive altruism.

Line 56-58: How do you define 'a species' sociobiology'? It would seem preferable to me to avoid vague language like this. Once more all of this is about the expression of condition-dependent facultative altruism with or without simultaneous direct fitness gains (captured by mutualistic cooperation) – nothing of what is here appears to challenge the monogamy hypothesis for the evolution of obligate reproductive altruism.

Line 59-63: Also this point seems misplaced as criticism of the monogamy hypothesis, because it

seems to represent a 'parasocial' arrangement, which the Boomsma reviews explicitly excluded as ancestral state for the evolution of superorganismality in the Wheeler sense.

Line 63-67: This makes sense to me, but it is perhaps premature to highlight here based on just a submitted paper, particularly because all of this seems consistent with the Hamiltonian paradigm that ecology (the b/c ration in Hamilton's rule) is as decisive as the relatedness terms when all three vary at the same time, as is always the case when evaluating cases of condition-dependent expression of facultative reproductive altruism. All the monogamy hypothesis did was claim that maximal and invariable relatedness was a necessary condition for making major evolutionary transitions to obligate reproductive altruism with permanently unmated workers – so for this kind of transitions relatedness comes first, cancels out of Hamilton's rule, so that $b/c > 1$ comes to represent the sufficient condition for actually making such transitions. A recent paper (Smith, Kent, Boomsma, Stow, *Nature Ecology and Evolution* 2018) shows that this proof of concept works fine to explain the evolution of life-time unmated workers in an ambrosia beetle – the commentary by Nick Davies summarizes that very well.

Line 67-69: I think this point has been made earlier in a Keller and Reeve model – all fine. It just has no direct bearing on the assumptions and predictions of the Boomsma monogamy hypothesis.

Line 70: 'Recently the monogamy debate has extended ...' ?? If you carefully read the Boomsma reviews all the way back to 2007 you will see that this was the core of the argument right from the start – even though the language was made more explicit in later reviews. Ref 10 confuses the evolution of first helpers at the nest with the point of no return to obligate reproductive altruism, which are very different things. Even though some others have also confused these issues in previous papers, you cannot hold the logic presented by Boomsma responsible for that confusion.

Line 75-76: No, of course not. But maternal manipulation is only a (partial) alternative explanation to voluntary reproductive altruism in social systems where some nests retain helpers and other nests do not. All those dynamics are covered by the classic dynamics of Hamilton's rule and have no bearing on the monogamy hypothesis as formulated for the evolution of life-time unmated nursing castes.

Line 76-87: For all I know, Davies & Gardner showed that the Olejarz et al. model was unsatisfactory because it worked with caricature single locus variation with extreme phenotypic effects. Ever since Alan Grafen wrote his well-known phenotypic gambit paper in the 1980s the field has been clear about complex social traits being largely independent of such single locus extreme effects. Making models of that kind (as Olejarz, Nonacs, and some others like to do), therefore has possible bearing on the first (facultative) expression of reproductive altruism, but not on the evolution of permanently sterile castes, which requires significant rewiring of complex gene-regulation networks depending on many genes with small effects. That is why social traits are best considered to have quantitative genetic variation with overall heritabilities that determine how such traits respond to selection, but without significant single-gene effects. I would think this should also apply to many cases of the expression of facultative helping when governed by the type of phenotypic plasticity proposed by Mary Jane West Eberhard. Also here it is hard to conceptualize how complex behavioural reaction norms are dependent on single locus variation with extreme phenotypic effects.

Line 98-100: I disagree with this blanket statement, because it suggests that all arguments have equal merits across the board. The Boomsma monogamy hypothesis has met with unambiguous support for the predictions that it actually made (major transitions to Wheeler superorganismality and to obligate multicellularity cf. Fisher, Cornwallis & West, *Current Biology* 2013). Other models, among them some by Nonacs, have shown that things are more

complex for facultative reproductive altruism. Under the assumptions stated, these models are likely correct, but they are not at variance with the monogamy hypothesis as originally framed. They shed light on cases of haplodiploid cooperative breeding and complement more general insights in (usually diploid) cooperative breeding in other arthropods and vertebrates. When taking that broader perspective, haplodiploidy is actually quite an idiosyncratic issue, because it implies that unmatedness does not necessarily imply sterility as it does in diploids. I think also that should be taken into account when evaluating the merits of a loser version of the monogamy hypothesis that has now been tested and found to be essentially correct in many diploid cooperative breeders (alpaeid shrimp, cooperatively breeding birds, cooperatively breeding mammals, etc – see recent Boomsma reviews for references).

Line 104-142: This list is likely to be in need of qualifications as current relevance seems doubtful.

1. Bourke & Franks (1995) argue that maternal manipulation is essentially a Hamiltonian concept, as was agreed upon by Richard Alexander, so any discussion of the validity of this idea should be framed in that context. Also manipulating mothers are under selection to maximize their inclusive fitness, so manipulating provisioning of daughters in their larval stages so they stay should not give mothers fewer grandchildren than provisioning them optimally so they can disperse and breed.
2. So far 'social heterosis' has never been shown to play a role empirically for facultative reproductive altruism; it has for obligate altruism in honeybees and leaf-cutting ants with obligate multiple queen mating. It seems crucial to acknowledge that distinction.
3. I do not think it is appropriate to capture humans and paper wasps under the same heading – overall none of these two systems seem relevant for the monogamy hypothesis as originally framed.
4. This seems misguided because the monogamy hypothesis is about obligate reproductive altruism (and in fact obligate multicellular soma as well) evolving 'voluntarily'. That is why maximal (invariable) relatedness is key because it makes the r -terms cancel out of Hamilton's rule. That makes the power issue brought up here moot. The Davies & Gardner model acknowledges that, any model coming out of Nowak's group does not.
5. As far as 'groups' are parasocial aggregations, this argument is irrelevant for the monogamy hypothesis which requires colony founding by a life-time monogamous pair.
6. Also this point seems hardly relevant, because outbreeding is normally secured in social systems. The few cases in which societies have offspring hanging around that do not obtain indirect fitness are irrelevant for the monogamy hypothesis. As Michael Griesser argued (recent PLOS Biology paper) there are forms of family life that essentially represent extended parental care. They will imply interesting social dynamics, but there is no relevance to the monogamy hypothesis when offspring do not help to obtain indirect fitness benefits.
- 7+8. Also these topics do not seem relevant for the monogamy hypothesis. They are of interest, but have been of interest ever since inclusive fitness theory was proposed so are hardly novel; antagonistic interlocus genetics remains poorly understood even in sexual selection research – it seems hardly relevant as a pressing research agenda here.

In sum, the title of the paper is too broad for the issues the paper actually evaluates. The monogamy hypothesis was never about cooperation for mutual direct fitness benefits, but only about the evolution of obligate reproductive altruism. Suggesting it was more by using 'cooperation' muddles the water. Many of these 1-8 topics are of interest, but just not in the context at hand.

Line 145-149: All valuable scientific theory is dogmatic. How else would Einstein and Bohr have succeeded replacing Newton mechanics? G.C. Williams explicitly emphasized the need for biological theory to also be stringent, so its predictions are vulnerable to falsification – that is why he explicitly endorsed inclusive fitness theory in his 1966 book *Adaptation and Natural Selection*. Neither I nor any others who have developed or tested the monogamy hypothesis have ever claimed it will be relevant for any social system. However, the framing has been increasingly precise about what is predicted and what is not (see particularly Boomsma reviews from 2013,

2014 and 2018), and these exclude scenarios involving the expression of facultative reproductive altruism with less than maximal sibling relatedness or cooperation for mutual direct benefits.

Line 149-153: As far as I am aware we know of no examples where 'emergent group level benefits' have been shown to challenge inclusive fitness theory. In general, any invocation of 'emergence' is unhelpful for scientific progress. As J.B.S. Haldane wrote: 'Particularly hostile to true scientific progress are the extremer forms of emergence [S]cience is committed to the attempt to unify human experience by explaining the complex in terms of the simple - The Causes of Evolution (1932). I think Peter Nonacs adheres to this principle when he closes off writing "Trust in inclusive fitness, but verify". However, it would then help for maintaining transparency in the literature if he would refrain from sociobiology terminology that was never intended to accurately capture the assumptions and predictions of inclusive fitness theory, and from using strawman versions of extensions of inclusive fitness theory to line up arguments that do not address real issues.

Decision letter (RSOS-180913.R0)

17-Sep-2018

Dear Dr Nonacs,

The editors assigned to your paper ("Monogamy both does and does not promote the evolution of altruistic cooperation") have now received comments from reviewers. We would like you to revise your paper in accordance with the referee and Associate Editor suggestions which can be found below (not including confidential reports to the Editor). Please note this decision does not guarantee eventual acceptance.

Please submit a copy of your revised paper before 10-Oct-2018. Please note that the revision deadline will expire at 00.00am on this date. If we do not hear from you within this time then it will be assumed that the paper has been withdrawn. In exceptional circumstances, extensions may be possible if agreed with the Editorial Office in advance. We do not allow multiple rounds of revision so we urge you to make every effort to fully address all of the comments at this stage. If deemed necessary by the Editors, your manuscript will be sent back to one or more of the original reviewers for assessment. If the original reviewers are not available, we may invite new reviewers.

- Data accessibility

<http://datadryad.org/submit?journalID=RSOS&manu=RSOS-180913>

- Competing interests

- Authors' contributions

- Acknowledgements

- Funding statement

Please note that Royal Society Open Science charge article processing charges for all new submissions that are accepted for publication. Charges will also apply to papers transferred to Royal Society Open Science from other Royal Society Publishing journals, as well as papers

submitted as part of our collaboration with the Royal Society of Chemistry (<http://rsos.royalsocietypublishing.org/chemistry>). If your manuscript is newly submitted and subsequently accepted for publication, you will be asked to pay the article processing charge, unless you request a waiver and this is approved by Royal Society Publishing. You can find out more about the charges at <http://rsos.royalsocietypublishing.org/page/charges>. Should you have any queries, please contact openscience@royalsociety.org.

on behalf of Dr Alexander Ophir (Associate Editor) and Prof. Kevin Padian (Subject Editor)
openscience@royalsociety.org

Subject Editor's comments:

Not surprisingly, there are strong differences here; I believe the reviewers will respect differences of analysis, but please make sure that you are getting the original literature correct, because they maintain that in several important places you are not (you may of course show them wrong). If you need more time than that allotted to revise, please let us know. Thanks and good luck.

Associate Editor's comments (Dr Alexander Ophir):

Associate Editor: 1

Comments to the Author:

Dear Dr. Nonacs,

I have received two reviews for your manuscript. It will come as no surprise that your paper has stirred passions, but both reviewers felt that the paper should have a chance for response and revision, and I agree. I look forward to receiving your thoughtful response to these comments.

Comments to Author:

Reviewers' Comments to Author:

Reviewer: 1

Comments to the Author(s)

In this comment, Prof. Nonacs gives his view of the inclusive fitness controversy and related issues, partly through the lens of two recent papers: Olejarz et al (2015, eLife) and Davies and Gardner (2018, Royal Society Open Science).

Olejarz et al (2015) analysed a model of the evolution of voluntary worker non-reproduction in haplodiploid insect societies and found that monogamy sometimes promotes and sometimes inhibits the evolution of worker sterility. Davies and Gardner (2018) analysed the same model and found that the results of Olejarz et al. were due to restrictive assumptions concerning the genetics of worker sterility. When we (Davies and Gardner) relaxed these assumptions, we found that monogamy actually has a clear promoting effect on the evolution of worker sterility.

I have to say that I don't agree entirely with Prof. Nonacs' interpretation of my paper, but I do

appreciate that he is trying to propose a way forward here and so I'll mainly restrict myself in this review to commenting on what I see as a few small matters of factual correctness. I don't think any of these problems are fundamental to what is being said in this comment piece more generally, but do need to be addressed.

Lines 82–84: "Davies and Gardner [24] expanded this analysis by allowing sterility alleles to be intermediate in their effect and beneficial to the queen in allowing her to gain more reproduction."

The first part of this sentence is right, but the second part doesn't seem right to me. Olejarz et al. made the assumption that the p_z function (proportion of male eggs produced by the queen) would be an increasing function of z . We adopted the same assumption. Therefore I don't think this can be described as a way in which we extended the analysis of Olejarz et al. To be clear, we did extend their analysis, but we did so by (1) allowing sterility alleles to be intermediate in their effect, (2) considering the long-term evolution of worker sterility rather than the short term invasibility of particular alleles, and (3) looking at a broader range of worker sterility scenarios.

On a related note, further up in this paragraph, it is stated that Olejarz et al. assumed that "the production of [worker-derived] males competes with and possibly reduces the number of males that are queen-produced." This is technically right but I think phrased in a way that could be clearer. Olejarz et al. made the assumption that queen male production increases with worker sterility, so I feel like the word "possibly" isn't quite right here. Overall production depends on both r_z and p_z , so you *could* get a situation where worker sterility causes everyone's production to decrease (decreasing r_z), which *could* reduce the queen's male production overall (depending on p_z), but I think overall the way this is described here might be a little confusing to readers.

Lines 92–94: Davies and Gardner [24] respond to Olejarz et al [12] by declaring that "monogamy always promotes... worker sterility" without qualifying that this requires traits to be expressed additively..."

We spend quite a lot of the paper exploring alternative scenarios in which worker sterility is dominant or recessive, so I don't know where this is coming from – we absolutely do not require traits to be expressed additively.

Lines 124–126: "The degree to which parents or siblings can impose complete or almost complete sterility on their group mates will determine whether the Olejarz et al. [12] or Davies and Gardner [24] models are more appropriate."

I'm not sure I understand this – we make the same assumptions as Olejarz et al when it comes to who is controlling sterility, so I don't think whether parents or siblings can impose sterility on their group mates has anything to do with whether our analysis is more appropriate than Olejarz et al.'s.

Finally, I wonder if the author would consider toning down the last paragraph just a little. I have always appreciated Prof. Nonacs' point of view, which has come to me through peer review a couple of times, and I appreciate that he makes his points in a measured way and does not seek to bury papers that he disagrees with. I also agree in general with the assessment in this paragraph that diversity of opinion can be fruitful and that excessive dogmatism is unhelpful. But I don't think I or my coauthor can fairly be described as people who consider "proving their opponents wrong" to be the most important thing, and it feels a bit like that is what is being said here. I worked on this paper during my PhD and I would never want to discourage anyone from wanting to study social evolution. In fact, I met with Carl Veller (one of the authors on Olejarz et

al, who was a PhD student like me at the time) in person to ask him for feedback on the paper, and because I didn't want the authors—especially the less senior ones—to feel like there was any bad blood between us. So although I'm sure this wasn't Prof. Nonacs' intention, the final paragraph feels a little bit unfair to me.

Best wishes,

Nick Davies

Reviewer: 2

Comments to the Author(s)

This manuscript evaluates the extent to which monogamy promotes altruistic cooperation. It is not completely clear to me why Peter Nonacs appears to feel that there are scientific reasons for considering possible alternatives to inclusive fitness theory for explaining patterns and processes of social evolution, which he does in practice in spite of his closing sentence. Apart from the reference to Nowak et al (2010) nothing in his own work (which appears in several hypothesis-ideas towards the end of the manuscript) seems to justify that. I therefore believe that much of this essay sees controversy and disagreement where there in fact is none, provided one is clear about terms and definitions and the ways in which they were messed up when sociobiology became influential.

In the four comments below I attempt to clarify this overall assessment. After that I provide specific comments with reference to line numbers:

1. Nowak et al. (2010) has nothing to do with the topic at hand. The reference to the Nowak et al. (2010) paper at the start of this manuscript is odd. That paper ignored the monogamy hypothesis (in spite of two Boomsma reviews of 2007 and 2009) having laid it out in considerable detail and the Science paper by Hughes et al. (2008) having shown it was supported by considerable empirical evidence. The Nowak et al. challenge of inclusive fitness theory was thoroughly refuted by a series of 5 rebuttals of which only one (Boomsma et al., 2011) explicitly addressed monogamy – the total number of authors involved was ca. 150 (not 134 as Nonacs states). The agenda of the Nowak et al. paper failed, because a scientific challenge of a major paradigm needs to achieve two things: 1. It needs to allow easy understanding of all insights already validated under the previous paradigm (in this case the 16 subfields listed with triple 'yes' in Table 1 of the Abbot et al. rebuttal that summarized the overwhelming empirical and modelling confirmation of inclusive fitness theory), and 2. Such a challenge needs to add a series of new testable predictions that were not valid questions under the existing paradigm, and that would have sufficient initial support to open up entire new research fields. This is what relativity theory (Einstein) and quantum theory (Bohr) did when they built on and enriched Newton mechanics. Nothing like that was achieved by Nowak et al. (2010). Their statement of 'meagre' support is simply wrong. Neither did their paper allow the recapture of any previous understanding or generate a single novel testable prediction. It therefore seems inappropriate to start the introduction of the present manuscript with Nowak et al. (2010). It would also be nice if Nonacs could actually mention more explicitly that the monogamy hypothesis has been very well supported by empirical data, not unexpectedly as it is in fact just an extension of Hamilton's rule and the inclusive fitness theory behind.

2. Nonacs' formulation of the central prediction of the monogamy hypothesis is incorrect. Nonacs appears to misrepresent what the monogamy hypothesis actually predicts. He is not the only one, because the Hughes et al. (Science 2008) paper testing the empirical evidence as provided by the then available comparative data is actually not very precise in how it phrased the idea being tested. If Nonacs is interested in evaluating the status of the monogamy hypothesis, he should

therefore go back to its exact formulation by Boomsma, both in the first review cited by Hughes et al. (*Current Biology* 2007) and reread the precise phrasing in the review paper that he cites (Boomsma 2009). That should reveal that the monogamy hypothesis was ONLY meant to explain the evolution of life-time physically differentiated castes, not the evolution of all altruistic cooperation as the title of the present Nonacs manuscript suggests. It was stated explicitly that this concerned the evolution of superorganismality *sensu* Wheeler, which Boomsma coined ‘obligate eusociality’ struggling with the ‘eusociality definition’ that Wilsonian sociobiology had left the field with. It has become increasingly clear (and has been expressed in increasingly explicit language by Boomsma (2009, 2013), Boomsma, Huszár & Pedersen (2014), and Boomsma & Gawne (2018) that sociobiology terminology is a smokescreen because its ‘eusociality’ does not discriminate between first helpers at the nest and life-time differentiated castes. The monogamy hypothesis never claimed to explain all aspects of what determined first helpers at the nest. It is about predicting the emergence of obligate altruism, not of facultative (condition-dependent) forms of altruism (facultative eusociality in the Boomsma reviews). Nonacs has made this interpretational mistake in his earlier modelling papers, which he now admits (e.g. nest inheritance), but his present text continues to repeat some of the same misapprehensions. His arguments thus acquire several strawman characteristics and they are already apparent in the title of the manuscript (the monogamy hypothesis is not about cooperation because reproductive altruism \neq cooperation).

3. Assumptions should be explicit and biologically realistic for models to be meaningful. It is of crucial importance for evaluating the assumptions that go into any model meant to describe the evolutionary origin of different types of altruistic phenotypes. The previous Nonacs models have always addressed the origin of helpers at the nest, not the origin of obligatory altruistic castes, i.e. the irreversible transition to Wheeler-superorganismality which implies that ALL colony members belong to a single adult caste phenotype for life. The monogamy hypothesis never intended to challenge the recurrent origins of facultative (condition-dependent) altruism through the simple logic of Hamilton’s rule ($rb > c$). If all three variables vary freely there can be situations where lower than full-sib relatedness still allowed some offspring to become helpers at the nest, for example because they were poorly endowed as adult phenotypes (maternal manipulation could play a role here, but note that mothers also need to weigh how they get most genes passed on to grand-offspring by making offspring disperse rather than stay). However, what the life-time monogamy hypothesis states is that it cannot be expected that this will ever lead to the evolution of permanently unmated workers (true workers as termite researchers aptly call them; neuters as Darwin and Wheeler conceptualized them). The monogamy hypothesis is about the evolution of obligate (unconditional) altruism, a distinction that is not made in the Hughes et al (2008) paper but that is explicit in the Boomsma reviews. Most examples that Nonacs highlights are about the origin of first helpers at the nest, and none is about helping behaviour going to fixation so that physically differentiated permanent castes appear. If Nonacs would make that distinction, most of the rationale for his critique would seem to disappear. The same rationale as outlined here also pervaded the recent Davies and Gardner paper, so the Nonacs critique appears to compare apples and oranges. A recent modelling paper by Quinones & Pen (*Nature Communications*, 2017) also shows clearly that strict monogamy is a necessary condition for the evolution of obligate, not facultative reproductive altruism. The Nonacs (2011, 2014) papers thus appear to have their assumptions wrong for them to be relevant as tests of the monogamy hypotheses for permanent castes – they address the evolution of first helpers.

4. The proposed ‘alternative’ hypotheses towards the end of the manuscript. Some of these appear to be interesting research agendas, but others seem rather far-fetched and lack more than hand-waving justification. More importantly, however, very few if any of them are directly relevant for testing the monogamy hypothesis in its original meaning.

I hope the following line-by-line comments will clarify these general points.

Line 26: 'eusociality' here lumps societies and superorganisms, which Boomsma & Gawne show are fundamentally different social systems, going back to Darwin, Weismann, Wheeler, Fisher, Williams and others all the way through the 1960s, until sociobiology started to water down and muddle these definitions. The Boomsma reviews (particularly 2013) also make this point, emphasizing that Crespi & Yanega (1995) and Beekman et al (2006) also criticized the 'eusociality continuum' ideas emanating from the watered down definitions by E.O. Wilson in the 1970s.

Line 29: Refs 9+10+12 had their modelling assumptions wrong for the monogamy hypothesis as originally formulated by Boomsma, which ref. 12 partly rectifies and the recent Davies&Gardner paper makes more explicit. You also admit that later on, but without stating the explicit reasons. There is also a semantic issue here. Empirical data can never be 'put to mathematical tests', not even in physics. Mathematical models can be used to derive predictions from axiomatic assumptions that can then be empirically tested, not the other way around. Being precise about this distinction is crucial particularly in evolutionary biology, where the main function of models is to make sense of empirically observed variation in social behaviours and life-histories - anything beyond that is illusionary.

Line 30: The monogamy hypothesis is not about the evolution of cooperation (for mutual benefits) but about the evolution of obligate reproductive altruism. What Hughes et al. (2008) and Cornwallis et al. (2010) tested was whether low parental promiscuity favors facultative reproductive reproductive altruism (with for social insects obligate reproductive altruism as a putative later extension). They never tested the hypothesis in its most stringent version originally proposed.

Line 32-33: When you conceptualize the evolution of permanent true worker castes as an evolutionary point of no return (i.e. a major evolutionary transition), as all Boomsma reviews since 2007 have done, the later transitions to multiple queen mating and polygyny (the latter primarily in ants - see Boomsma, Huszár & Pedersen, *Animal Behaviour* 2014) are not relevant for the argument, because by then true worker castes could no longer go back to facultative (condition-dependent) altruism. Also here 'cooperative behaviour' is an inaccurate description of what the monogamy hypothesis is about (same in Line 35).

Line 35-40: Once more, the monogamy hypothesis is not about these cases. If daughters stay to help even though they could have dispersed, there is no reason to question that they do that because the inequality specified by Hamilton's rule compels them to do so. Because these cases represent facultative altruism, it is expected that measured average direct and indirect fitnesses are in fact equal (representing a frequency dependent ESS equilibrium where both strategies do equally well). This is what the cited studies report and it is all consistent with inclusive fitness theory, even though the papers try to argue otherwise - very little if anything in these decision making processes can be shown to be maladaptive or challenging established inclusive fitness theory (line 41-42). But the main issue is that none of this challenges the monogamy hypothesis for its prediction of major evolutionary transitions to superorganismality as defined by Wheeler (Boomsma & Gawne, 2018).

Line 44: The above implies there is no dissonance between theory and empirical results as implied here, provided one phrases the predictions and assumptions of the Boomsma hypothesis correctly, rather than forcing them into uninformative sociobiology jargon that Boomsma tried to avoid.

Line 50: Again, the argument is not about cooperation but about reproductive altruism.

Line 56-58: How do you define 'a species' sociobiology'? It would seem preferable to me to avoid

vague language like this. Once more all of this is about the expression of condition-dependent facultative altruism with or without simultaneous direct fitness gains (captured by mutualistic cooperation) – nothing of what is here appears to challenge the monogamy hypothesis for the evolution of obligate reproductive altruism.

Line 59-63: Also this point seems misplaced as criticism of the monogamy hypothesis, because it seems to represent a ‘parasocial’ arrangement, which the Boomsma reviews explicitly excluded as ancestral state for the evolution of superorganismality in the Wheeler sense.

Line 63-67: This makes sense to me, but it is perhaps premature to highlight here based on just a submitted paper, particularly because all of this seems consistent with the Hamiltonian paradigm that ecology (the b/c ration in Hamilton’s rule) is as decisive as the relatedness terms when all three vary at the same time, as is always the case when evaluating cases of condition-dependent expression of facultative reproductive altruism. All the monogamy hypothesis did was claim that maximal and invariable relatedness was a necessary condition for making major evolutionary transitions to obligate reproductive altruism with permanently unmated workers – so for this kind of transitions relatedness comes first, cancels out of Hamilton’s rule, so that $b/c > 1$ comes to represent the sufficient condition for actually making such transitions. A recent paper (Smith, Kent, Boomsma, Stow, *Nature Ecology and Evolution* 2018) shows that this proof of concept works fine to explain the evolution of life-time unmated workers in an ambrosia beetle – the commentary by Nick Davies summarizes that very well.

Line 67-69: I think this point has been made earlier in a Keller and Reeve model – all fine. It just has no direct bearing on the assumptions and predictions of the Boomsma monogamy hypothesis.

Line 70: ‘Recently the monogamy debate has extended ...’ ?? If you carefully read the Boomsma reviews all the way back to 2007 you will see that this was the core of the argument right from the start – even though the language was made more explicit in later reviews. Ref 10 confuses the evolution of first helpers at the nest with the point of no return to obligate reproductive altruism, which are very different things. Even though some others have also confused these issues in previous papers, you cannot hold the logic presented by Boomsma responsible for that confusion.

Line 75-76: No, of course not. But maternal manipulation is only a (partial) alternative explanation to voluntary reproductive altruism in social systems where some nests retain helpers and other nests do not. All those dynamics are covered by the classic dynamics of Hamilton’s rule and have no bearing on the monogamy hypothesis as formulated for the evolution of life-time unmated nursing castes.

Line 76-87: For all I know, Davies & Gardner showed that the Olejarz et al. model was unsatisfactory because it worked with caricature single locus variation with extreme phenotypic effects. Ever since Alan Grafen wrote his well-known phenotypic gambit paper in the 1980s the field has been clear about complex social traits being largely independent of such single locus extreme effects. Making models of that kind (as Olejarz, Nonacs, and some others like to do), therefore has possible bearing on the first (facultative) expression of reproductive altruism, but not on the evolution of permanently sterile castes, which requires significant rewiring of complex gene-regulation networks depending on many genes with small effects. That is why social traits are best considered to have quantitative genetic variation with overall heritabilities that determine how such traits respond to selection, but without significant single-gene effects. I would think this should also apply to many cases of the expression of facultative helping when governed by the type of phenotypic plasticity proposed by Mary Jane West Eberhard. Also here it is hard to conceptualize how complex behavioural reaction norms are dependent on single locus variation with extreme phenotypic effects.

Line 98-100: I disagree with this blanket statement, because it suggests that all arguments have equal merits across the board. The Boomsma monogamy hypothesis has met with unambiguous support for the predictions that it actually made (major transitions to Wheeler superorganismality and to obligate multicellularity cf. Fisher, Cornwallis & West, *Current Biology* 2013). Other models, among them some by Nonacs, have shown that things are more complex for facultative reproductive altruism. Under the assumptions stated, these models are likely correct, but they are not at variance with the monogamy hypothesis as originally framed. They shed light on cases of haplodiploid cooperative breeding and complement more general insights in (usually diploid) cooperative breeding in other arthropods and vertebrates. When taking that broader perspective, haplodiploidy is actually quite an idiosyncratic issue, because it implies that unmatedness does not necessarily imply sterility as it does in diploids. I think also that should be taken into account when evaluating the merits of a loser version of the monogamy hypothesis that has now been tested and found to be essentially correct in many diploid cooperative breeders (alpaeid shrimp, cooperatively breeding birds, cooperatively breeding mammals, etc – see recent Boomsma reviews for references).

Line 104-142: This list is likely to be in need of qualifications as current relevance seems doubtful. 1. Bourke & Franks (1995) argue that maternal manipulation is essentially a Hamiltonian concept, as was agreed upon by Richard Alexander, so any discussion of the validity of this idea should be framed in that context. Also manipulating mothers are under selection to maximize their inclusive fitness, so manipulating provisioning of daughters in their larval stages so they stay should not give mothers fewer grandchildren than provisioning them optimally so they can disperse and breed. 2. So far ‘social heterosis’ has never been shown to play a role empirically for facultative reproductive altruism; it has for obligate altruism in honeybees and leaf-cutting ants with obligate multiple queen mating. It seems crucial to acknowledge that distinction. 3. I do not think it is appropriate to capture humans and paper wasps under the same heading – overall none of these two systems seem relevant for the monogamy hypothesis as originally framed. 4. This seems misguided because the monogamy hypothesis is about obligate reproductive altruism (and in fact obligate multicellular soma as well) evolving ‘voluntarily’. That is why maximal (invariable) relatedness is key because it makes the r -terms cancel out of Hamilton’s rule. That makes the power issue brought up here moot. The Davies & Gardner model acknowledges that, any model coming out of Nowak’s group does not. 5. As far as ‘groups’ are parasocial aggregations, this argument is irrelevant for the monogamy hypothesis which requires colony founding by a life-time monogamous pair. 6. Also this point seems hardly relevant, because outbreeding is normally secured in social systems. The few cases in which societies have offspring hanging around that do not obtain indirect fitness are irrelevant for the monogamy hypothesis. As Michael Griesser argued (recent *PLOS Biology* paper) there are forms of family life that essentially represent extended parental care. They will imply interesting social dynamics, but there is no relevance to the monogamy hypothesis when offspring do not help to obtain indirect fitness benefits. 7+8. Also these topics do not seem relevant for the monogamy hypothesis. They are of interest, but have been of interest ever since inclusive fitness theory was proposed so are hardly novel; antagonistic interlocus genetics remains poorly understood even in sexual selection research – it seems hardly relevant as a pressing research agenda here.

In sum, the title of the paper is too broad for the issues the paper actually evaluates. The monogamy hypothesis was never about cooperation for mutual direct fitness benefits, but only about the evolution of obligate reproductive altruism. Suggesting it was more by using ‘cooperation’ muddles the water. Many of these 1-8 topics are of interest, but just not in the context at hand.

Line 145-149: All valuable scientific theory is dogmatic. How else would Einstein and Bohr have succeeded replacing Newton mechanics? G.C. Williams explicitly emphasized the need for

biological theory to also be stringent, so its predictions are vulnerable to falsification – that is why he explicitly endorsed inclusive fitness theory in his 1966 book *Adaptation and Natural Selection*. Neither I nor any others who have developed or tested the monogamy hypothesis have ever claimed it will be relevant for any social system. However, the framing has been increasingly precise about what is predicted and what is not (see particularly Boomsma reviews from 2013, 2014 and 2018), and these exclude scenarios involving the expression of facultative reproductive altruism with less than maximal sibling relatedness or cooperation for mutual direct benefits.

Line 149-153: As far as I am aware we know of no examples where ‘emergent group level benefits’ have been shown to challenge inclusive fitness theory. In general, any invocation of ‘emergence’ is unhelpful for scientific progress. As J.B.S. Haldane wrote: ‘Particularly hostile to true scientific progress are the extremer forms of emergence [S]cience is committed to the attempt to unify human experience by explaining the complex in terms of the simple - *The Causes of Evolution* (1932). I think Peter Nonacs adheres to this principle when he closes off writing “Trust in inclusive fitness, but verify”. However, it would then help for maintaining transparency in the literature if he would refrain from sociobiology terminology that was never intended to accurately capture the assumptions and predictions of inclusive fitness theory, and from using strawman versions of extensions of inclusive fitness theory to line up arguments that do not address real issues.

Author's Response to Decision Letter for (RSOS-180913.R0)

See Appendix A.

RSOS-180913.R1 (Revision)

Review form: Reviewer 1 (Nicholas Davies)

Is the manuscript scientifically sound in its present form?

Yes

Are the interpretations and conclusions justified by the results?

Yes

Is the language acceptable?

Yes

Is it clear how to access all supporting data?

Not Applicable

Do you have any ethical concerns with this paper?

No

Have you any concerns about statistical analyses in this paper?

No

Recommendation?

Accept with minor revision (please list in comments)

Comments to the Author(s)

As before, I am limiting myself to commenting upon issues of factual correctness, especially regarding Davies and Gardner (2018), rather than issues of interpretation or opinion. That's not to say that I think natural history is a matter of opinion. As the author states, people might disagree on how to define eusociality or any other word, but I don't agree that this is merely a "philosophical" issue because some definitions increase our understanding and others don't.

Because I'm an author of the piece that is being commented upon, I think this is the appropriate role to take here and although I don't agree with everything being said here, my feeling is that that debate is better held in the open.

Lines 26-29. I think the relatedness term in Hamilton's rule is incorrectly defined here, because it seems like c is the number of foregone offspring for the actor, and b is the number of additional offspring for the recipient, so then r should be the relatedness between the actor and the recipient, not the relatedness between the actor and the recipient's offspring. (Or alternatively, c needs to be multiplied by the relatedness to the actor's offspring.)

Line 40. This is just a comment. In my view, there are two (related) reasons why the cited models don't conform with Hamilton's rule ($rb > c$). One is that some of the studies assume that altruistic phenotypes are encoded by alleles of large effect. I hope it is becoming increasingly clear to theorists and practitioners that predictions made from models that assume quantitative genetics or which make equivalent assumptions such as additivity of allelic effects (as Hamilton did in deriving $rb > c$; also see previous comment on quantitative genetics from reviewer 2) cannot simply be transposed into a model that assumes alleles of large effect. A lot of the disagreement over inclusive fitness really boils down to this issue. My own view is that the quantitative view is the more general one, since it is equivalent to the assumption that mutations of intermediate effect will eventually arise in a population. It also has empirical support for the reasons mentioned by reviewer 2. Finally, in cases where the predictions made by quantitative genetics and the predictions made by biallelic models diverge the most, that is precisely when there will be the strongest selection for an allele of intermediate effect to spread. So I really think there is good reason to believe that, over evolutionary time, populations will tend to conform to predictions made by quantitative-genetic models rather than models with alleles of fixed effect.

The second reason is that Hamilton's rule – in the strict sense of $rb > c$ – is designed to analyse a specific social phenotype, in which an actor gives up some of their reproductive fitness in order to increase the reproductive fitness of a relative and there are *no other impacts to their actions*. One can't take $rb > c$ and apply it unthinkingly when there are other things going on, like male production by workers, nest inheritance, and so on. The way that Gardner, myself and others apply inclusive fitness theory is by first building a model which (hopefully) captures the relevant phenomenon, then interpreting the results in terms of inclusive fitness, not by starting with $rb > c$ and trying to fit it to every social evolutionary problem that involves altruism. For example, Davies and Gardner (2018) don't end up with $rb > c$, but a more complex condition.

Line 83. Is there a word missing before "evolution"?

Line 127. halictid typo.

Line 181. My middle initial is G, not D.

Best wishes

Nick Davies

Review form: Reviewer 2 (Jacobus J. Boomsma)

Is the manuscript scientifically sound in its present form?

No

Are the interpretations and conclusions justified by the results?

No

Is the language acceptable?

No

Is it clear how to access all supporting data?

Not Applicable

Do you have any ethical concerns with this paper?

No

Have you any concerns about statistical analyses in this paper?

No

Recommendation?

Major revision is needed (please make suggestions in comments)

Comments to the Author(s)

I appreciate the efforts by the author to clarify a number of points that I raised in my first review, which has given a better balance overall. However, my comments below will emphasize that further revision would be desirable if this paper is to serve the authors' purpose of pointing towards a way forward.

General comments:

- It remains puzzling to me that the author seems to think he can discuss the merits of monogamy as a key explanatory variable without precisely referring to where the idea comes from and how it developed. Before Boomsma (2007) the concept simply did not exist, except as a few throwaway sentences by Hamilton, Dawkins and Cronin (the introduction of that first review has the citations). If you choose a title like the one you have now, and aim to make a contribution that could ease a perceived controversy, you will have to relate to the concept as it was developed between 2007 and 2018 and at least briefly mention the consistent empirical evidence for it. This mentioning should differentiate between the strict monogamy (window) hypothesis for major transitions to permanent castes, and monogamy as reduced promiscuity serving as statistical predictor of the expression of context-dependent (facultative) altruism. These two aspects are really different issues and not acknowledging that difference muddles the water.

- If you did (re)read the full spectrum of monogamy reviews, you would notice that monogamy was ONLY claimed to have been a universal necessary condition (and NOT a sufficient condition) for the evolution of life-time unmated castes. It has also been made clear from the start (and in increasingly precise language in later reviews) that in all societies of cooperative breeders (e.g. halictids, Polistes, lower termites, naked mole rats, snapping shrimp, meerkats, family living birds) the three parameters of Hamilton's rule always vary simultaneously (even when relatedness is high). This implies that the monogamy hypothesis had nothing fundamentally new to say about those social systems where multiple (often most) co-breeding individuals have independent agency (totipotency) – the hypothesis only claimed to parsimoniously explain the

handful of convergent major transitions to obligate eusociality (Wheeler superorganismality; Boomsma & Gawne, 2018) where that independent agency has irreversibly disappeared. If you write (line 19) that 'No single topic better exemplifies the ongoing controversy than the question of what role does monogamy play in the evolution of reproductive division of labor within cooperating family associations', you just have to be explicit that you address monogamy as it applies to societies of cooperative breeders, not the strict major transitions monogamy-window hypothesis.

- With this in mind, I remain unsure whether I understand your title 'Hamilton's rule is essential but insufficient for understanding monogamy's role in social evolution'. I think we agree on the first part, but I do not understand the ending. The strict monogamy-window hypothesis is just a reduction of Hamilton's rule (the r terms cancel) and if monogamy is not 100% the standard version of Hamilton's rule continues to capture everything as a first-principle theory. So what is insufficient in your title supposed to mean? We know that inclusive fitness is the sum of direct fitness and indirect fitness, so do you mean that direct fitness benefits are often essential as long as altruism remains facultative? But that is true by definition according to Hamilton's rule. Even if mothers would manipulate daughters against their interests (for which the evidence is weak), parental manipulation does not represent something alien to inclusive fitness theory. Bourke and Franks (1995) have some pages of text explaining this. If this essay is to meant to objectively map out what we (dis)agree on and what is perhaps subject to debate, I think you should make clear that the monogamy (window) hypothesis predicted how major transitions to permanent castes have come about and then add that this may have confused readers to believe monogamy was an essential condition for all forms of social evolution. You could then proceed outlining why this is not the case and how/why Hamiltonian logic allows a variety of social traits to be maintained either for reasons of cooperation (no relatedness involved) or by facultative expression of altruism, or (perhaps) by offspring being manipulated into maximizing their mother's fitness. Anything else will be perceived as a broadside against the monogamy hypothesis as a whole that lacks the necessary nuance and will thus not achieve the balance you appear to aim for.

- Pitting the monogamy idea against NTW (2010), a paper that failed to make even a single testable prediction and that kept all its assumptions implicit, seems a tall order when the goal is to remain balanced. Modelling in ecology and evolution is ultimately about explaining variation in social systems of very different kinds from the same parsimonious set of first principles. Precise definitions are part of the transparent overall approach needed to achieve that - not merely a philosophical 'human imposed' issue. Evolutionary biology cannot be a serious science if one focuses on 'process' without having precise language to capture process, as the author argues in one of his responses. This contrast is particularly strange because NTW only modelled full sib colonies, so they implicitly worked with strict monogamy as your ref 15 pointed out. I still believe you do your mission a disservice by starting to argue from NTW (2010) in implicitly supportive language, because these authors were never interested in a proper discussion whereas you are.

- Assumptions of models do matter crucially (as the author highlights in line 63), but this arguments deserves to be generalized. In evolutionary biology models should both make their assumptions and predictions explicit, which is rarely consistently done for phenotypes in population genetic models. Furthermore, both assumptions and predictions should be testable. One cannot expect that tests of the predictions have been done when a model is novel, but scientific transparency requires that modellers tell their readers how their assumptions apply to specific groups of organisms and provide reasonable evidence that they are likely to be valid in a sufficient number of cases to make the model worth the trouble. The NTW paper is a case in point where essentially all crucial assumptions remained implicit and sometimes turned out nonsensical when digging them out (e.g. the idea that workers do not need to be considered as evolutionary agents). Another, already mentioned, implicit assumption in that paper was that

they in fact only considered full sib families where relatedness was constant and non-variable. I find it hard to imagine that you endorse of practices like this and believe you have sufficient experience as a field naturalist to agree that matching model assumptions with empirical evidence is a point of key importance.

- The disagreement issue that you address in this essay is real, but not a discussion between parties holding equally strong decks of cards - less than a handful empirically inclined biologists have seen any merit in NTW (2010), about 30 times fewer than the ones who signed the 2011 rebuttals. This is why I think you do your mission a disservice by focusing on that particular paper in the approving sense that your wording suggests. The issue is really about types of models to be used and the assumptions that feed them. I think the author and I agree that the significance of inclusive fitness theory has been immense. What we also should agree upon is the 'make assumptions explicit' issue summarized above. What (only in part) separates us is that he appears to feel that specific population genetic models have the potential to overturn general first-principle theory such as Hamilton's rule, where I feel they will only complement first principle theory. But at least here we have an issue that may well not be obvious to many biologists, so is worth clarifying because it never appears authors like Nowak, Van Veelen, Allen and E.O. Wilson have an interest in engaging in this discussion. My overall impression is that it always turns out that when Nowak's group produces an explicitly genetic model, it is just a matter of time until someone else (e.g. Nick Davies and Andy Gardner is the most recent case) shows those 'overturning conclusions' are simply a consequence of making rather extreme assumptions about how genes affect phenotypes. But that type of discussion is not new - Hamilton and later Dave Queller, Steve Frank, Stu West, Andy Gardner, James Marshall, Peter Taylor, Francois Rousset, etc have all made it clear that strong selection, drift, dominance, non-additivity, etc will make Hamilton's rule only approximately valid or sometimes make it fail. To see why, one should realize that as soon as one goes down the path of more specific 'realistic' models the extra assumptions needed (relative to the Price-equation inclusive fitness approach) make such models less general. As Dave Queller writes in the Introduction of his 2017 AmNat paper: 'Levins (1966) discussed how models have to trade off between generality, realism and precision. Kuhn (1977) similarly noted that theories face conflicts between accuracy, consistency, scope, simplicity', and fruitfulness. It is my clear impression that the author agrees with this but it remains very implicit.

- The non-novelty of the NTW critique on the Price equation approach has been aptly described by Steve Frank, a mathematical scholar on par with Nowak. If the author would care to check his series of seven papers in the *Journal of Evolutionary Biology* (2011-2013), he would find all the references to these early insights. Particularly Frank (2012) Natural selection. IV. The Price equation (JEB 25, 1002-1019) is illuminating, because the final sections address the so called 'controversy'. I quote: 'This quote from van Veelen et al. (2012) demonstrates an interesting approach to scholarship. They first cite Frank as stating that dynamic insufficiency is a drawback of the Price equation. They then disagree with that point of view and present as their own interpretation an argument that is nearly identical in concept and phrasing to my own statement in the very paper that they cited as the foundation for their disagreement.' Given this kind of arguing, does it then surprise the author of the present essay that little appreciation for the approach of NTW (2010) has accumulated? Another quote that the author may find interesting is from Frank's final paper in this series entitled Natural selection. VII. History and interpretation of kin selection theory (JEB 26, 1151-1184, 2013) which says: 'Nowak et al. (2010) say that one must make a specific model for a specific case, and then one gets the right answer. True, but the history of science shows unambiguously that one gains a lot by understanding the abstract causal principles that join different cases and different models within a common framework (Frank, 2012b).' The upshot is, as Hamilton already knew, that explicit population genetic models are important, but they are (by definition) never general and generate few testable predictions.

- In light of the above, I retain my opinion that, for this essay to be influential, you need to make explicit that the role of monogamy as a predictor is crucially different in social domains where altruism is facultative (i.e. cooperative breeders) and domains where altruism is obligate (all helpers remain unmated; hence they are true workers - what Darwin meant with 'neuters'). Boomsma & Gawne (2018) have the most recent and complete update of that logic. Making this distinction is important because it seems futile you challenge the evidence for the major evolutionary transition logic of the monogamy hypothesis given the consistent evidence (the Smith et al. paper on *Austroplatypus ambrosia* beetles in *Nature E&E* (2018) gives a complementary summary of what Boomsma & Gawne reviewed). You can then focus on the cooperative breeders where becoming a helper remains facultative, so that either social nests coexist with solitary nests (e.g. *Megalopta*) or where reproductive success always requires colonial sociality but where relatedness is never high enough to create a 'monogamy window' (e.g. *Polistes*, naked mole rats, Seychelles warblers, meerkats). Focusing on that type of social systems is useful because your summary will then make clear that: 1. Monogamy may help, but not necessarily does help (e.g. *Megalopta* if maternal manipulation is real; greater anis because they do not live in family groups so indirect fitness is moot). 2. You will be able to clarify the misunderstanding that all social systems required a monogamy window - although never intended by Boomsma, this is how some people have misunderstood the phylogenies in Hughes et al. (2008). 3. That approach will put your own models in their right context for example when making clear that monogamy is not necessarily good if there is nest inheritance. 4. It will also allow connections with reproductive skew logic, which is about cooperative breeders that co-breed in spite of being totipotent (skew theory is not about permanent castes and obligate division of labor). 5. You might briefly refer to a discussion between Charlie Cornwallis & co, Dustin Rubinstein & co, and Michael Griesser & co on what drives cooperative breeding in birds. You will notice that what seemed a fundamental disagreement (relatedness first versus ecology first) now appears to be evaporating because Michael Griesser talks about extended parental care (no indirect fitness as in greater anis), and Cornwallis and Rubinstein recently published a joint paper agreeing on (as a broad trend) monogamy (low promiscuity) having promoted the colonization of harsh environments, but that living and radiating in harsh environments induced secondary reversals to promiscuity, some of that related to 'family life' giving way to groups of non-relatives. 6. In fact, all the six points towards the end of your paper are about cooperative breeding (none is about obligate worker castes) so all you need to do is make that explicit. 7. It would get you away from unfruitful discussions on whether inclusive fitness logic explains everything. Maternal manipulation or not, the key point is that helpers always obtain non-trivial indirect fitness benefits when they invest in helping (as they do in your refs 17-20), which is all that matters when altruism is expressed in a facultative (context dependent) manner. It would seem to me that an emphasis in this direction would make your essay much more valuable because you could show that: 1. There is no fundamental disagreement between almost everybody that different types of models serve complementary purposes. 2. That different types of models only do that when previous insights are respected (provided they were thoroughly documented - NTW did not show that respect), and 3. That monogamy in the sense of low promiscuity is a statistical facilitator for cooperative breeding, not a necessary condition as it apparently was for evolving obligate 'neuter' castes or, as termite researchers would say, for evolving 'true workers'.

- I find much of this so called 'controversy' frustrating, because I do not feel responsible for it although I am happy to take credit for the monogamy hypothesis. If the author would care to re-read the closing paragraph of Boomsma (2009) I hope he can see why. I would now not have phrased this closing statement in sociobiology 'eusociality language', but by contrasting Wheeler-superorganismality (obligate unmated altruism) against cooperative breeding (among totipotent reproductive agents expressing at best facultative altruism) as done in later reviews. However, the meaning remains the same. The cooperative breeder domain of social evolution is more

taxonomically diverse, more dynamic, and more variable in causation. Both models and empirical evidence show that and it is by acknowledging that distinction that a more transparent version of your essay can make a positive difference.

Specific points:

Line 16. 'arguably intensified' - I think this phrasing needs adjustment. To my knowledge no new authors have joined the small group of inclusive fitness theory critics. So we would need citations here proving that statement.

Line 17. 'aggressively vociferous' - This kind of language should be avoided. Overall, it would be sufficient if lines 15-17 merely stated that no consensus has been reached. Who started making this debate acrimonious is open to interpretation. As I explained above, most opinionated evolutionary biologists considered NTW (2010) to be an unduly aggressive and poorly argued attempt to discredit the most successful general theory of adaptation produced in the 20th century.

Line 33. To keep the discussion general, I think this should say 'as from multiple mating or cooperative breeding of multiple females'

Line 39. 'touted' is unnecessary pejorative. Another case where language moderation is in order. The statement also seems to be formally incorrect. The 'monogamy window' is a theoretical concept (Boomsma, 2007, 2009, 3013) that has only been claimed to be a necessary condition for major transitions to coloniality with permanent castes (see general comments above).

Line 42. 'above phylogenies'? - you mean phylogenetically controlled comparative analyses?

Line 45. 'for unexplained reasons'?? It has been known for many years (Boomsma & Ratnieks, 1996; Strassmann, 2001) that obligate multiple queen mating in ants and corbiculate bees is a secondary derived evolutionary development, later confirmed to also apply to the vespine wasps by the studies of Foster and Ratnieks, all well before Hughes et al. (2008). The monogamy hypothesis provides a simple parsimonious general explanation for these convergent phenomena by stipulating that once the point of no return to irreversibly unmated worker castes had been passed, these secondary developments were free to evolve when they provided lasting benefits. If workers no longer have the plasticity to opt out by mating and dispersing, then one should not expect any group disharmony to arise except as a secondary consequence of chimeric colonies (requiring for example worker policing as documented most thoroughly in the honey bee). This point has been made in a single discussion sentence by Hamilton (1964), somewhat more elaborately by Boomsma & Ratnieks (1996), and explicitly in all the monogamy review published since 2007. That not all later empirical papers have been precise on this point cannot be held against the hypothesis having been transparent. It is actually at this level that the authors 'heterosis' hypothesis works.

Line 47-55. Terms like 'cast doubt' and 'very maladaptive' are inconsistent with the evidence and would thus best be phrased more carefully. This section claims there is repeated evidence for worker behaviour that does not make sense from an inclusive fitness perspective. I have checked all four studies and cannot see how they provide strong evidence for the inference that Hamiltonian logic is problematic. Nonacs et al. (2006) tested reproductive skew models that the author later dismissed as having limited general value (Nonacs & Hager, 2011; Biological Reviews). Skew models are derived from inclusive fitness logic but failed to live up to expectations because they needed too many special assumptions that could hardly be tested. It is thus no great surprise that the qualitative cross-taxa predictions in the 2006 study appeared to work, but the detailed population-level predictions did not. The second study (Rehan et al., 2014)

is about a bee where two sisters may build joint nests and where indirect fitness of subordinates was only half the direct fitness of nesting alone. However, that study did not measure the mortality costs of dispersing and initiating a nest alone. If that cost would be twice the failure rate of staying as a helper, the two options would be equivalent and consistent with Hamilton's rule (as a mixed ESS not unlike the birds in your ref 14). The Gadagkar (2016) paper is essentially a review of decades of work and ends with the conclusion: 'Clearly, Hamilton's rule provides a powerful framework for understanding the evolution of social and altruistic behaviour in *R. marginata*. Even though I have often complained that most investigators focus excessively or exclusively on r and neglect the b and c terms in Hamilton's rule, the power of the theory is evident when all three factors are considered, as I have attempted to do for *R. marginata*.' Kapheim et al. (2015) is the best and most recent single case study, but also here authors had to infer what direct fitness helpers could have had in a rather indirect way. This allowed them to quantify productivity alternatives, but they could not consider the possibility that dispersal rather than helping would likely incur significantly higher mortality costs. In sum, none of these studies refute inclusive fitness logic because helpers obtained measurable inclusive fitness benefits in all cases. Such qualitative confirmations are often the best that can be achieved, because quantifying the Hamiltonian b and c variables remains essentially impossible in natural settings where dispersal costs are almost impossible to obtain. I suggest the author adjusts the phrasing of this paragraph to do justice to what was really shown. As far as I can see not all authors of these four papers have invoked maternal manipulation and Kapheim et al. only inferred that mechanism was consistent with the data. It has not been proven.

Line 56. Having reached this point it is really hard to see the 'dissonance'. In all cases where tests were done (your refs 13, 14, 17-20) non-trivial indirect fitness benefits were always found, consistent with Hamiltonian logic. If there were quantitative mismatches (the indirect fitness benefits seeming not to be substantial enough), the most parsimonious explanation was that the non-measured costs of dispersal were substantial.

Line 68-70. Yes of course, but this is the key argument of the monogamy window hypothesis for the emergence of life-time unmated castes. Under strict monogamy the r -terms cancel out of Hamilton's rule, so all that matters is the $b > c$ condition - i.e. what you call life-history and ecology. Before lineages reach that point of no-return what matters is $br > c$. This may seem a subtle difference, but it is not, because many combinations of b and r can make the inequality work. You just need substantial benefits applying in particular nest settings (and not in others) to express conditional altruism even when relatedness is modest. But altruism will never become obligate in such scenarios. That apparently only happened when relatedness to siblings remained identical to relatedness to offspring over a huge number of generations, i.e. a sufficient number to irreversibly rewire complete developmental pathways (with hideously complex gene regulation networks) to produce permanent queen and worker castes. The assumption of a worker taking over the nest from her dead mother has never been part of the predictive domain of the monogamy window hypothesis. It is clearly a derived trait in lineages where permanent castes evolved, and as far as nest inheritance happens in societies with facultative altruism it may well prevent they can reach the monogamy window needed to evolve permanent castes. I believe that is consistent with your models, but phrasing this precisely matters for transparency of argument.

Line 75-76. The point here is whether maternal manipulation can induce irreversible transitions to life-time true-worker castes. There is no evidence to my knowledge that this has been modelled or observed to happen. Also the halictid systems considered in the Quinones & Pen model ultimately have to converge on strict monogamy before true helper traits can go to fixation (i.e. show the first signs of becoming irreversible).

Line 78. 'also considered'? unclear phrasing. You mean 'considered a scenario'?

Line 82-87. I think this summary of the Davies and Gardner paper is understating the clarity of their results to a degree that does not serve the balanced view the author says he pursues. The first sentence will never communicate to the unprepared reader that their analyses showed that none of the conclusions of the Olejarz model really held up when realistic population genetics were used. The second sentence is totally non-transparent and can easily be read as the Davies and Gardner paper having unresolved loose ends, which I do not think was the case within the framework they considered. I also remember the Davies and Gardner paper showing the argument of worker sons not to hold up in their analyses, so more clear language (and references) would be needed if you disagree.

Line 89. Once more – the original monogamy hypothesis was primarily about the evolution of life-time unmated castes in the ants, bees, wasps and termites. Only after making that first-principle inference did Boomsma (2007) suggest that similar tendencies might be occurring in cooperative breeders, i.e. make the expression of facultative altruism more likely. It was that latter part that was tested and found to be correct in cooperatively breeding birds and mammals (Cornwallis; Lukas). Your phrasing is so general that this key point is glossed over, which muddles the water. In cooperative breeders monogamy is a statistical predictor; for the evolution of true workers it becomes a necessary condition. These are very different things (see general comments above).

Line 90-91. ‘enhance, retard, or have no effect ...’ as ‘overall conclusion’. Of what, of models that did not make their assumptions explicit? I am not aware of any empirical data set that has shown that the monogamy hypothesis was incorrect if you consider (as you should) that it never intended to replace Hamilton’s rule in vertebrate cooperative breeders and social insects where altruism is facultative (condition dependent).

Line 92-95. This is true, but it seems to be just restating Hamilton’s rule. No monogamy paper every claimed to replace that logic.

Line 96-105. My compliments for the constructive language in this paragraph. This is what the tone should be throughout this essay. You might want to add some of the 2010-2011 papers by Van Veelen as good examples of acrimonious inclusive fitness critique.

Line 106-113. It is unclear to me why the tone of this paragraph then becomes unbalanced again. I think ‘prosecutorial’ is once more a type of language to be avoided. All these papers did was reason from first GENERAL principles against a paper that was broadly conceived as a very unreasonable attack on massive progress in general understanding of social evolution based on inclusive fitness theory (see my general intro comments). It was NTW who started ignoring all the evidence (which you acknowledge in the previous paragraph). Objections were not ‘disingenuous’ (again wording that should be removed). All we can say is that there is disagreement about whether single-locus modelling with strong discontinuous effects on assumed phenotypes is useful or not (see my general comment above). Whether the mathematics was correct is besides the point. The equations were correct in all papers as far as I know, else criticism would have been much stronger. The point is that when assumptions are wrong, the outcome will make no sense. In such cases results are not inconvenient (another value laden term to be reconsidered), but irrelevant for what happens in natural populations. Please rephrase this paragraph to achieve balanced coverage as you do in the preceding paragraph.

Line 112. ‘... exceptions accumulate’ I am not aware of such accumulation. Do you mean the same four papers addressed above (refs 17-20)? You would have to give new references to justify ‘accumulate’.

Line 113. ‘uncritical’? To my knowledge almost every test of inclusive fitness theory has been an independent challenge of the general applicability of first-principle inclusive fitness theory. Critical attitudes in the Popperian sense have thus abounded and the theory has survived these refutation assaults at least in a qualitative sense. You may find that having more quantitative matches would be desirable, and I would agree. However, even the qualitative matches have been inspiring enough to generate new avenues of derived inclusive fitness theory, as for example David Haig’s genomic imprinting theory. By the way, also Dave Queller and Steve Frank have always argued that both types of models are useful and complementary.

Line 114. Remove the value-laden word like ‘presciently’

Line. 118-121. NTW did not pioneer simulation modelling and simulation modelling is something else than formal mathematical modelling. Formal mathematics provides first principle theory such as Hamilton’s rule and its broadened version using the Price equation. Simulation models address specific cases where multiple restrictive assumptions need to be made, and where complexity exceeds what can be tracked analytically. As I tried to explain in my general comments above, simulation approaches can be useful but only if the assumptions about what phenotypes do are reasonable, and they should never claim to demonstrate something truly general. Overselling in that way was typical for the NTW paper and because their assumptions were often wrong or irrelevant, it gained no followers. I do not understand why the author is so adamant to defend this paper as some kind of a ‘flagship case’. It detracts from the merits of his own models, which are more realistic and where he is willing to be specific about assumptions and about matches with general theory.

Line 124-125. ‘could consider’. Yes, but again the power issue is fundamentally different for lineages where colonies have facultative altruism versus those that have obligate altruism. You should make explicit that your point is about the former, not the latter type of social systems, and cite the well-known review by Beekman and Ratnieks to build any new argument on that synthesis. In fact it turns out that not only this first point, but also the following points 2-6 are about social systems with facultative altruism (see my last general comment above).

Line 136. ‘can easily select’ As far as I am aware there is no evidence for this to be ‘easy’ and to my knowledge there are no empirical data showing that selection for genetic diversity has produced consistent group adaptations in cooperative breeders. Reference 41 reports patterns across empirical studies but without addressing whether drifting ‘workers’ are potential reproducers or not. The only exceptions of unambiguous group adaptations owing to higher (chimeric) genetic diversity within groups appear to have happened after the point of no return to superorganismality had been passed. I.e., honeybee and leaf-cutting ant colonies deal better with disease when they have more patriline and costly obligate multiple queen mating is therefore a functional adaptation. Whether drifting workers can be considered to be a group adaptation remains to be proven and seems unlikely from a general theoretical perspective – these phenomena are merely effects of something else that could easily not have a functional adaptive significance (see Williams, 1966, for an important discussion about function and effect). Referring to ‘indirect reciprocity’ is unhelpful if you do not add a few lines of explanation of how that concept connects to what this essay is about.

Line 140-141. The monogamy hypothesis has always emphasized life-time (obligate) unmatedness, not obligate sterility, which requires diploidy to be an automatic consequence of unmatedness.

Line 144-146. This is a hand-waving statement. The monogamy hypothesis predicts (both in the early reviews and more explicitly in the later ones, which you should cite if you aim to be

balanced) that such reversals should not have happened. That prediction has been consistent with all available evidence so far (see Boomsma & Gawne 2018 and Smith et al. 2018 for updates), so a statement like this is misleading unless you can cite empirical papers that have provided evidence to the contrary. A more productive approach would be to make sure you acknowledge that the 'monogamy window' hypothesis for major transitions to caste societies is something else than facilitating monogamy in cooperative breeders (see my general comments) – and focus on the latter.

Line 147. The monogamy (window) hypothesis is about how facultative altruism can be replaced by obligate altruism, not about cooperation.

Line 148-150. Yes, it should work both ways, but only if relatedness is a true variable as in cooperatively breeding vertebrates, *Polistes* wasps, halictid bees etc. Once more, the core idea of the monogamy hypothesis is that strict full-sibling relatedness makes the relatedness terms cancel. That was apparently a necessary condition for major irreversible transitions to higher organismal complexity (both colonial superorganismality and obligate multicellular organismality), but not a sufficient one (the Leggett et al. paper – your ref 7 – also makes this point). Only long-term consistent $b > c$ benefits will forge major irreversible transitions after the necessary condition is fulfilled and that is a tall order. This entire point 3 is fine if you make explicit it is about cooperative breeders only (just like the other 5 points). In such 'societies' monogamy is never strict. References 4 and 8 are perfectly up front about that. Reference 3 was not, but you cannot hold that against the monogamy hypothesis. It is easy to plot the major irreversible transitions on the Hughes et al (2008) phylogeny (the base of the corbiculate bees, the base of the higher termites, the base of the vespine wasps). What you say here applies basal to those transitions, not beyond them – the later monogamy reviews have been very explicit about that, but that logic is also in the 2007 and 2009 versions. Irreversibility of transitions to coloniality where all individuals belong to a complementary caste for life (Wheeler superorganismality) is not just an assumption, it is consistent with all comparative data both for superorganismality and obligate multicellularity (cf Smith et al. *Nature E&E* 2018, Fisher et al. *Current Biology* 2013, Boomsma & Gawne, 2018).

Line 151-156. Yes, *Polistes* colonies and human groups are societies, albeit very different ones because we have cumulative culture driving our social evolution. So again this is a 'society' argument, but it has no bearing on the strict type of monogamy window hypothesized to represent a necessary condition for the emergence of obligate altruism in superorganisms (which no longer are societies of totipotent individuals). Once more, it would be useful if you somehow made these distinctions explicit when you intend this essay to be a way forward.

Line 158. 'sometimes'? Sibling competition within groups is generally and principally important as Hamilton and May (1977), Steve Frank (JEB 2012), Stu West & Ido Pen, and Leggett et al. (your ref 7) have amply documented. The group benefits you have in mind here are just the Hamiltonian $b > c$ values. Also your point 5 is about family groups before permanent castes evolve.

Line 162. And so is point 6. The Riehl work is very interesting, but not as a challenge of the monogamy hypothesis for altruism driven by indirect fitness benefits, because two key conditions do not apply: 1. 'helpers' gain no inclusive fitness – they just hang around to receive prolonged parental investment. 2. These greater ants are not family groups where altruism could have evolved.

Line 168-172. The final quote is nice, but should in my opinion refer to the need to test both the assumptions and the predictions of any model used to better understand social evolution. It has no specific bearing on ecology and group-level benefits, which are fully covered by Hamilton's

rule as the series of Steve Frank papers in JEB (2011-2013) amply document. Does it have bearings on genetics? Yes, quite possibly so, but remember Hamilton's rule is about genetics as well. If you want to make the genetics more explicit that can be useful, but it will always imply making restrictive assumptions that the Price equation (general inclusive fitness) approach does not need to make. If you keep adding special assumptions in order to gain 'realism', you may end up with a model that is very precise about a scenario that may never apply in nature. See my general comments and the Queller quote given above on how generality and precision of models trade-off. Both approaches are valuable when assumptions about individual and colony phenotypes are reasonable - and then they are complementary ways towards understanding. Models with inappropriate or unspecified assumptions are not useful. I think that should be a major condition for making your 'verify' by modelling credible, but the final arbiter of course remains what the empirical data show.

Koos Boomsma, Department of Biology, University of Copenhagen, Denmark

<https://orcid.org/0000-0002-3598-1609>

Decision letter (RSOS-180913.R1)

19-Nov-2018

Dear Dr Nonacs:

Manuscript ID RSOS-180913.R1 entitled "Hamilton's rule is essential but insufficient for understanding monogamy's role in social evolution" which you submitted to Royal Society Open Science, has been reviewed. The comments of the reviewer(s) are included at the bottom of this letter.

Please submit a copy of your revised paper before 12-Dec-2018. Please note that the revision deadline will expire at 00.00am on this date. If we do not hear from you within this time then it will be assumed that the paper has been withdrawn. In exceptional circumstances, extensions may be possible if agreed with the Editorial Office in advance. We do not allow multiple rounds of revision so we urge you to make every effort to fully address all of the comments at this stage. If deemed necessary by the Editors, your manuscript will be sent back to one or more of the original reviewers for assessment. If the original reviewers are not available we may invite new reviewers.

- Ethics statement

- Data accessibility

- Competing interests

- Authors' contributions

- Acknowledgements

- Funding statement

Please note that Royal Society Open Science charge article processing charges for all new submissions that are accepted for publication. Charges will also apply to papers transferred to Royal Society Open Science from other Royal Society Publishing journals, as well as papers submitted as part of our collaboration with the Royal Society of Chemistry (<http://rsos.royalsocietypublishing.org/chemistry>). If your manuscript is newly submitted and subsequently accepted for publication, you will be asked to pay the article processing charge, unless you request a waiver and this is approved by Royal Society Publishing. You can find out

more about the charges at <http://rsos.royalsocietypublishing.org/page/charges>. Should you have any queries, please contact openscience@royalsociety.org.

on behalf of Dr Alexander Ophir (Associate Editor) and Professor Kevin Padian (Subject Editor)
openscience@royalsociety.org

Associate Editor Comments to Author (Dr Alexander Ophir):

Associate Editor: 1

Comments to the Author:

Dear Dr. Nonacs,

Thank you for your revised manuscript. I have received evaluations from the two reviewers who kindly agreed to review your manuscript a second time. Both have signed their reviews. Your paper continues to stir strong opinions and disagreement but I think you continue to raise an interesting perspective that deserves a place in the record that will be open to and inspire continued debate.

Because the motivation to write your paper was based on a response to his paper that is now published in RSOS, Reviewer 1 (Dr. Davies) has limited his comments to identifying typos and evaluating the facts presented. He has admirably provided you room for you to argue your interpretations of them even where he disagrees. The few remaining comments he provides should be easily addressed or adjusted.

Reviewer 2 (Dr. Boomsma) has provided a thoughtful critique of your revision. I was pleased to see that he felt that your revision was much improved. Based on my reading of this review, I believe the focus of his current critique falls to the following: Elements of your manuscript require more explicit definitions for terms that you discuss, and more precision in your handling of the domains of social evolution that you are considering. He was puzzled by the structure of your essay and felt you might re-structure it to do yourself and your position a greater service. I think you should carefully consider this constructive feedback and that you can evaluate how to best structure your essay. Although you should not weaken your views when there is good support for them, the reviewer has identified a few places where potentially unnecessarily polarizing language could be toned down to make your paper more accessible to others where appropriate.

I think that you should consider this round of comments and adjust your manuscript where appropriate, and defend your position in other cases. I strongly commend you for attempting to shine light on a topic that merits attention, despite the potentially strong feelings of disagreement on the topic. I commend the reviewers for serving as fact checkers while also allowing you to argue a different perspective. It is my personal opinion that this is ultimately the best of what science represents: a forum for active and impassioned disagreement in pursuit of accuracy, truth, and new discovery. Doing so with grace and cordiality is a criterion that we should all hold

ourselves to, and I think you and the reviewers are doing a good job of living up to that mark. Thank you all. I look forward to receiving what I hope will be a final round of edits from you soon.

Best
Alex Ophir

Associate Editor: 2
Comments to the Author:
Dear Dr. Nonacs,

Thank you for your thoughtful and balanced revision. I will now invite the reviewers to consider your responses to their comments.

Best
Alex Ophir

Reviewer comments to Author:
Reviewer: 1

Comments to the Author(s)

As before, I am limiting myself to commenting upon issues of factual correctness, especially regarding Davies and Gardner (2018), rather than issues of interpretation or opinion. That's not to say that I think natural history is a matter of opinion. As the author states, people might disagree on how to define eusociality or any other word, but I don't agree that this is merely a "philosophical" issue because some definitions increase our understanding and others don't.

Because I'm an author of the piece that is being commented upon, I think this is the appropriate role to take here and although I don't agree with everything being said here, my feeling is that that debate is better held in the open.

Lines 26-29. I think the relatedness term in Hamilton's rule is incorrectly defined here, because it seems like c is the number of foregone offspring for the actor, and b is the number of additional offspring for the recipient, so then r should be the relatedness between the actor and the recipient, not the relatedness between the actor and the recipient's offspring. (Or alternatively, c needs to be multiplied by the relatedness to the actor's offspring.)

Line 40. This is just a comment. In my view, there are two (related) reasons why the cited models don't conform with Hamilton's rule ($rb > c$). One is that some of the studies assume that altruistic phenotypes are encoded by alleles of large effect. I hope it is becoming increasingly clear to theorists and practitioners that predictions made from models that assume quantitative genetics or which make equivalent assumptions such as additivity of allelic effects (as Hamilton did in deriving $rb > c$; also see previous comment on quantitative genetics from reviewer 2) cannot simply be transposed into a model that assumes alleles of large effect. A lot of the disagreement over inclusive fitness really boils down to this issue. My own view is that the quantitative view is the more general one, since it is equivalent to the assumption that mutations of intermediate effect will eventually arise in a population. It also has empirical support for the reasons mentioned by reviewer 2. Finally, in cases where the predictions made by quantitative genetics and the predictions made by biallelic models diverge the most, that is precisely when there will be the strongest selection for an allele of intermediate effect to spread. So I really think there is

good reason to believe that, over evolutionary time, populations will tend to conform to predictions made by quantitative-genetic models rather than models with alleles of fixed effect.

The second reason is that Hamilton's rule – in the strict sense of $rb > c$ – is designed to analyse a specific social phenotype, in which an actor gives up some of their reproductive fitness in order to increase the reproductive fitness of a relative and there are *no other impacts to their actions*. One can't take $rb > c$ and apply it unthinkingly when there are other things going on, like male production by workers, nest inheritance, and so on. The way that Gardner, myself and others apply inclusive fitness theory is by first building a model which (hopefully) captures the relevant phenomenon, then interpreting the results in terms of inclusive fitness, not by starting with $rb > c$ and trying to fit it to every social evolutionary problem that involves altruism. For example, Davies and Gardner (2018) don't end up with $rb > c$, but a more complex condition.

Line 83. Is there a word missing before "evolution"?

Line 127. halictid typo.

Line 181. My middle initial is G, not D.

Best wishes

Nick Davies

Reviewer: 2

Comments to the Author(s)

I appreciate the efforts by the author to clarify a number of points that I raised in my first review, which has given a better balance overall. However, my comments below will emphasize that further revision would be desirable if this paper is to serve the authors' purpose of pointing towards a way forward.

General comments:

- It remains puzzling to me that the author seems to think he can discuss the merits of monogamy as a key explanatory variable without precisely referring to where the idea comes from and how it developed. Before Boomsma (2007) the concept simply did not exist, except as a few throwaway sentences by Hamilton, Dawkins and Cronin (the introduction of that first review has the citations). If you choose a title like the one you have now, and aim to make a contribution that could ease a perceived controversy, you will have to relate to the concept as it was developed between 2007 and 2018 and at least briefly mention the consistent empirical evidence for it. This mentioning should differentiate between the strict monogamy (window) hypothesis for major transitions to permanent castes, and monogamy as reduced promiscuity serving as statistical predictor of the expression of context-dependent (facultative) altruism. These two aspects are really different issues and not acknowledging that difference muddles the water.
- If you did (re)read the full spectrum of monogamy reviews, you would notice that monogamy was ONLY claimed to have been a universal necessary condition (and NOT a sufficient condition) for the evolution of life-time unmated castes. It has also been made clear from the start (and in increasingly precise language in later reviews) that in all societies of cooperative breeders (e.g. halictids, Polistes, lower termites, naked mole rats, snapping shrimp, meerkats, family living birds) the three parameters of Hamilton's rule always vary

simultaneously (even when relatedness is high). This implies that the monogamy hypothesis had nothing fundamentally new to say about those social systems where multiple (often most) co-breeding individuals have independent agency (totipotency) – the hypothesis only claimed to parsimoniously explain the handful of convergent major transitions to obligate eusociality (Wheeler superorganismality; Boomsma & Gawne, 2018) where that independent agency has irreversibly disappeared. If you write (line 19) that ‘No single topic better exemplifies the ongoing controversy than the question of what role does monogamy play in the evolution of reproductive division of labor within cooperating family associations’, you just have to be explicit that you address monogamy as it applies to societies of cooperative breeders, not the strict major transitions monogamy-window hypothesis.

- With this in mind, I remain unsure whether I understand your title ‘Hamilton’s rule is essential but insufficient for understanding monogamy’s role in social evolution’. I think we agree on the first part, but I do not understand the ending. The strict monogamy-window hypothesis is just a reduction of Hamilton’s rule (the r terms cancel) and if monogamy is not 100% the standard version of Hamilton’s rule continues to capture everything as a first-principle theory. So what is insufficient in your title supposed to mean? We know that inclusive fitness is the sum of direct fitness and indirect fitness, so do you mean that direct fitness benefits are often essential as long as altruism remains facultative? But that is true by definition according to Hamilton’s rule. Even if mothers would manipulate daughters against their interests (for which the evidence is weak), parental manipulation does not represent something alien to inclusive fitness theory. Bourke and Franks (1995) have some pages of text explaining this. If this essay is to meant to objectively map out what we (dis)agree on and what is perhaps subject to debate, I think you should make clear that the monogamy (window) hypothesis predicted how major transitions to permanent castes have come about and then add that this may have confused readers to believe monogamy was an essential condition for all forms of social evolution. You could then proceed outlining why this is not the case and how/why Hamiltonian logic allows a variety of social traits to be maintained either for reasons of cooperation (no relatedness involved) or by facultative expression of altruism, or (perhaps) by offspring being manipulated into maximizing their mother’s fitness. Anything else will be perceived as a broadside against the monogamy hypothesis as a whole that lacks the necessary nuance and will thus not achieve the balance you appear to aim for.

- Pitting the monogamy idea against NTW (2010), a paper that failed to make even a single testable prediction and that kept all its assumptions implicit, seems a tall order when the goal is to remain balanced. Modelling in ecology and evolution is ultimately about explaining variation in social systems of very different kinds from the same parsimonious set of first principles. Precise definitions are part of the transparent overall approach needed to achieve that – not merely a philosophical ‘human imposed’ issue. Evolutionary biology cannot be a serious science if one focuses on ‘process’ without having precise language to capture process, as the author argues in one of his responses. This contrast is particularly strange because NTW only modelled full sib colonies, so they implicitly worked with strict monogamy as your ref 15 pointed out. I still believe you do your mission a disservice by starting to argue from NTW (2010) in implicitly supportive language, because these authors were never interested in a proper discussion whereas you are.

- Assumptions of models do matter crucially (as the author highlights in line 63), but this arguments deserves to be generalized. In evolutionary biology models should both make their assumptions and predictions explicit, which is rarely consistently done for phenotypes in population genetic models. Furthermore, both assumptions and predictions should be testable. One cannot expect that tests of the predictions have been done when a model is novel, but scientific transparency requires that modellers tell their readers how their assumptions apply to specific groups of organisms and provide reasonable evidence that they are likely to be valid in a

sufficient number of cases to make the model worth the trouble. The NTW paper is a case in point where essentially all crucial assumptions remained implicit and sometimes turned out nonsensical when digging them out (e.g. the idea that workers do not need to be considered as evolutionary agents). Another, already mentioned, implicit assumption in that paper was that they in fact only considered full sib families where relatedness was constant and non-variable. I find it hard to imagine that you endorse of practices like this and believe you have sufficient experience as a field naturalist to agree that matching model assumptions with empirical evidence is a point of key importance.

- The disagreement issue that you address in this essay is real, but not a discussion between parties holding equally strong decks of cards - less than a handful empirically inclined biologists have seen any merit in NTW (2010), about 30 times fewer than the ones who signed the 2011 rebuttals. This is why I think you do your mission a disservice by focusing on that particular paper in the approving sense that your wording suggests. The issue is really about types of models to be used and the assumptions that feed them. I think the author and I agree that the significance of inclusive fitness theory has been immense. What we also should agree upon is the 'make assumptions explicit' issue summarized above. What (only in part) separates us is that he appears to feel that specific population genetic models have the potential to overturn general first-principle theory such as Hamilton's rule, where I feel they will only complement first principle theory. But at least here we have an issue that may well not be obvious to many biologists, so is worth clarifying because it never appears authors like Nowak, Van Veelen, Allen and E.O. Wilson have an interest in engaging in this discussion. My overall impression is that it always turns out that when Nowak's group produces an explicitly genetic model, it is just a matter of time until someone else (e.g. Nick Davies and Andy Gardner is the most recent case) shows those 'overturning conclusions' are simply a consequence of making rather extreme assumptions about how genes affect phenotypes. But that type of discussion is not new - Hamilton and later Dave Queller, Steve Frank, Stu West, Andy Gardner, James Marshall, Peter Taylor, Francois Rousset, etc have all made it clear that strong selection, drift, dominance, non-additivity, etc will make Hamilton's rule only approximately valid or sometimes make it fail. To see why, one should realize that as soon as one goes down the path of more specific 'realistic' models the extra assumptions needed (relative to the Price-equation inclusive fitness approach) make such models less general. As Dave Queller writes in the Introduction of his 2017 AmNat paper: 'Levins (1966) discussed how models have to trade off between generality, realism and precision. Kuhn (1977) similarly noted that theories face conflicts between accuracy, consistency, scope, simplicity', and fruitfulness. It is my clear impression that the author agrees with this but it remains very implicit.

- The non-novelty of the NTW critique on the Price equation approach has been aptly described by Steve Frank, a mathematical scholar on par with Nowak. If the author would care to check his series of seven papers in the Journal of Evolutionary Biology (2011-2013), he would find all the references to these early insights. Particularly Frank (2012) Natural selection. IV. The Price equation (JEB 25, 1002-1019) is illuminating, because the final sections address the so called 'controversy'. I quote: 'This quote from van Veelen et al. (2012) demonstrates an interesting approach to scholarship. They first cite Frank as stating that dynamic insufficiency is a drawback of the Price equation. They then disagree with that point of view and present as their own interpretation an argument that is nearly identical in concept and phrasing to my own statement in the very paper that they cited as the foundation for their disagreement.' Given this kind of arguing, does it then surprise the author of the present essay that little appreciation for the approach of NTW (2010) has accumulated? Another quote that the author may find interesting is from Frank's final paper in this series entitled Natural selection. VII. History and interpretation of kin selection theory (JEB 26, 1151-1184, 2013) which says: 'Nowak et al. (2010) say that one must make a specific model for a specific case, and then one gets the right answer. True, but the history of science shows unambiguously that one gains a lot by understanding the abstract causal

principles that join different cases and different models within a common framework (Frank, 2012b).’ The upshot is, as Hamilton already knew, that explicit population genetic models are important, but they are (by definition) never general and generate few testable predictions.

- In light of the above, I retain my opinion that, for this essay to be influential, you need to make explicit that the role of monogamy as a predictor is crucially different in social domains where altruism is facultative (i.e. cooperative breeders) and domains where altruism is obligate (all helpers remain unmated; hence they are true workers - what Darwin meant with ‘neuters’). Boomsma & Gawne (2018) have the most recent and complete update of that logic. Making this distinction is important because it seems futile you challenge the evidence for the major evolutionary transition logic of the monogamy hypothesis given the consistent evidence (the Smith et al. paper on *Austroplatypus ambrosia* beetles in *Nature E&E* (2018) gives a complementary summary of what Boomsma & Gawne reviewed). You can then focus on the cooperative breeders where becoming a helper remains facultative, so that either social nests coexist with solitary nests (e.g. *Megalopta*) or where reproductive success always requires colonial sociality but where relatedness is never high enough to create a ‘monogamy window’ (e.g. *Polistes*, naked mole rats, Seychelles warblers, meerkats). Focusing on that type of social systems is useful because your summary will then make clear that: 1. Monogamy may help, but not necessarily does help (e.g. *Magalopta* if maternal manipulation is real; greater anis because they do not live in family groups so indirect fitness is moot). 2. You will be able to clarify the misunderstanding that all social systems required a monogamy window – although never intended by Boomsma, this is how some people have misunderstood the phylogenies in Hughes et al. (2008). 3. That approach will put your own models in their right context for example when making clear that monogamy is not necessarily good if there is nest inheritance. 4. It will also allow connections with reproductive skew logic, which is about cooperative breeders that co-breed in spite of being totipotent (skew theory is not about permanent castes and obligate division of labor). 5. You might briefly refer to a discussion between Charlie Cornwallis & co, Dustin Rubinstein & co, and Michael Griesser & co on what drives cooperative breeding in birds. You will notice that what seemed a fundamental disagreement (relatedness first versus ecology first) now appears to be evaporating because Michael Griesser talks about extended parental care (no indirect fitness as in greater anis), and Cornwallis and Rubinstein recently published a joint paper agreeing on (as a broad trend) monogamy (low promiscuity) having promoted the colonization of harsh environments, but that living and radiating in harsh environments induced secondary reversals to promiscuity, some of that related to ‘family life’ giving way to groups of non-relatives. 6. In fact, all the six points towards the end of your paper are about cooperative breeding (none is about obligate worker castes) so all you need to do is make that explicit. 7. It would get you away from unfruitful discussions on whether inclusive fitness logic explains everything. Maternal manipulation or not, the key point is that helpers always obtain non-trivial indirect fitness benefits when they invest in helping (as they do in your refs 17-20), which is all that matters when altruism is expressed in a facultative (context dependent) manner. It would seem to me that an emphasis in this direction would make your essay much more valuable because you could show that: 1. There is no fundamental disagreement between almost everybody that different types of models serve complementary purposes. 2. That different types of models only do that when previous insights are respected (provided they were thoroughly documented – NTW did not show that respect), and 3. That monogamy in the sense of low promiscuity is a statistical facilitator for cooperative breeding, not a necessary condition as it apparently was for evolving obligate ‘neuter’ castes or, as termite researchers would say, for evolving ‘true workers’.

- I find much of this so called ‘controversy’ frustrating, because I do not feel responsible for it although I am happy to take credit for the monogamy hypothesis. If the author would care to re-read the closing paragraph of Boomsma (2009) I hope he can see why. I would now not have phrased this closing statement in sociobiology ‘eusociality language’, but by contrasting Wheeler-

superorganismality (obligate unmated altruism) against cooperative breeding (among totipotent reproductive agents expressing at best facultative altruism) as done in later reviews. However, the meaning remains the same. The cooperative breeder domain of social evolution is more taxonomically diverse, more dynamic, and more variable in causation. Both models and empirical evidence show that and it is by acknowledging that distinction that a more transparent version of your essay can make a positive difference.

Specific points:

Line 16. 'arguably intensified' - I think this phrasing needs adjustment. To my knowledge no new authors have joined the small group of inclusive fitness theory critics. So we would need citations here proving that statement.

Line 17. 'aggressively vociferous' - This kind of language should be avoided. Overall, it would be sufficient if lines 15-17 merely stated that no consensus has been reached. Who started making this debate acrimonious is open to interpretation. As I explained above, most opinionated evolutionary biologists considered NTW (2010) to be an unduly aggressive and poorly argued attempt to discredit the most successful general theory of adaptation produced in the 20th century.

Line 33. To keep the discussion general, I think this should say 'as from multiple mating or cooperative breeding of multiple females'

Line 39. 'touted' is unnecessary pejorative. Another case where language moderation is in order. The statement also seems to be formally incorrect. The 'monogamy window' is a theoretical concept (Boomsma, 2007, 2009, 3013) that has only been claimed to be a necessary condition for major transitions to coloniality with permanent castes (see general comments above).

Line 42. 'above phylogenies'? - you mean phylogenetically controlled comparative analyses?

Line 45. 'for unexplained reasons'?? It has been known for many years (Boomsma & Ratnieks, 1996; Strassmann, 2001) that obligate multiple queen mating in ants and corbiculate bees is a secondary derived evolutionary development, later confirmed to also apply to the vespine wasps by the studies of Foster and Ratnieks, all well before Hughes et al. (2008). The monogamy hypothesis provides a simple parsimonious general explanation for these convergent phenomena by stipulating that once the point of no return to irreversibly unmated worker castes had been passed, these secondary developments were free to evolve when they provided lasting benefits. If workers no longer have the plasticity to opt out by mating and dispersing, then one should not expect any group disharmony to arise except as a secondary consequence of chimeric colonies (requiring for example worker policing as documented most thoroughly in the honey bee). This point has been made in a single discussion sentence by Hamilton (1964), somewhat more elaborately by Boomsma & Ratnieks (1996), and explicitly in all the monogamy review published since 2007. That not all later empirical papers have been precise on this point cannot be held against the hypothesis having been transparent. It is actually at this level that the authors 'heterosis' hypothesis works.

Line 47-55. Terms like 'cast doubt' and 'very maladaptive' are inconsistent with the evidence and would thus best be phrased more carefully. This section claims there is repeated evidence for worker behaviour that does not make sense from an inclusive fitness perspective. I have checked all four studies and cannot see how they provide strong evidence for the inference that Hamiltonian logic is problematic. Nonacs et al. (2006) tested reproductive skew models that the author later dismissed as having limited general value (Nonacs & Hager, 2011; Biological

Reviews). Skew models are derived from inclusive fitness logic but failed to live up to expectations because they needed too many special assumptions that could hardly be tested. It is thus no great surprise that the qualitative cross-taxa predictions in the 2006 study appeared to work, but the detailed population-level predictions did not. The second study (Rehan et al., 2014) is about a bee where two sisters may build joint nests and where indirect fitness of subordinates was only half the direct fitness of nesting alone. However, that study did not measure the mortality costs of dispersing and initiating a nest alone. If that cost would be twice the failure rate of staying as a helper, the two options would be equivalent and consistent with Hamilton's rule (as a mixed ESS not unlike the birds in your ref 14). The Gadagkar (2016) paper is essentially a review of decades of work and ends with the conclusion: 'Clearly, Hamilton's rule provides a powerful framework for understanding the evolution of social and altruistic behaviour in *R. marginata*. Even though I have often complained that most investigators focus excessively or exclusively on r and neglect the b and c terms in Hamilton's rule, the power of the theory is evident when all three factors are considered, as I have attempted to do for *R. marginata*.' Kapheim et al. (2015) is the best and most recent single case study, but also here authors had to infer what direct fitness helpers could have had in a rather indirect way. This allowed them to quantify productivity alternatives, but they could not consider the possibility that dispersal rather than helping would likely incur significantly higher mortality costs. In sum, none of these studies refute inclusive fitness logic because helpers obtained measurable inclusive fitness benefits in all cases. Such qualitative confirmations are often the best that can be achieved, because quantifying the Hamiltonian b and c variables remains essentially impossible in natural settings where dispersal costs are almost impossible to obtain. I suggest the author adjusts the phrasing of this paragraph to do justice to what was really shown. As far as I can see not all authors of these four papers have invoked maternal manipulation and Kapheim et al. only inferred that mechanism was consistent with the data. It has not been proven.

Line 56. Having reached this point it is really hard to see the 'dissonance'. In all cases where tests were done (your refs 13, 14, 17-20) non-trivial indirect fitness benefits were always found, consistent with Hamiltonian logic. If there were quantitative mismatches (the indirect fitness benefits seeming not to be substantial enough), the most parsimonious explanation was that the non-measured costs of dispersal were substantial.

Line 68-70. Yes of course, but this is the key argument of the monogamy window hypothesis for the emergence of life-time unmated castes. Under strict monogamy the r -terms cancel out of Hamilton's rule, so all that matters is the $b > c$ condition – i.e. what you call life-history and ecology. Before lineages reach that point of no-return what matters is $br > c$. This may seem a subtle difference, but it is not, because many combinations of b and r can make the inequality work. You just need substantial benefits applying in particular nest settings (and not in others) to express conditional altruism even when relatedness is modest. But altruism will never become obligate in such scenarios. That apparently only happened when relatedness to siblings remained identical to relatedness to offspring over a huge number of generations, i.e. a sufficient number to irreversibly rewire complete developmental pathways (with hideously complex gene regulation networks) to produce permanent queen and worker castes. The assumption of a worker taking over the nest from her dead mother has never been part of the predictive domain of the monogamy window hypothesis. It is clearly a derived trait in lineages where permanent castes evolved, and as far as nest inheritance happens in societies with facultative altruism it may well prevent they can reach the monogamy window needed to evolve permanent castes. I believe that is consistent with your models, but phrasing this precisely matters for transparency of argument.

Line 75-76. The point here is whether maternal manipulation can induce irreversible transitions to life-time true-worker castes. There is no evidence to my knowledge that this has been modelled or observed to happen. Also the halictid systems considered in the Quinones & Pen model

ultimately have to converge on strict monogamy before true helper traits can go to fixation (i.e. show the first signs of becoming irreversible).

Line 78. 'also considered'? unclear phrasing. You mean 'considered a scenario'?

Line 82-87. I think this summary of the Davies and Gardner paper is understating the clarity of their results to a degree that does not serve the balanced view the author says he pursues. The first sentence will never communicate to the unprepared reader that their analyses showed that none of the conclusions of the Olejarz model really held up when realistic population genetics were used. The second sentence is totally non-transparent and can easily be read as the Davies and Gardner paper having unresolved loose ends, which I do not think was the case within the framework they considered. I also remember the Davies and Gardner paper showing the argument of worker sons not to hold up in their analyses, so more clear language (and references) would be needed if you disagree.

Line 89. Once more – the original monogamy hypothesis was primarily about the evolution of life-time unmated castes in the ants, bees, wasps and termites. Only after making that first-principle inference did Boomsma (2007) suggest that similar tendencies might be occurring in cooperative breeders, i.e. make the expression of facultative altruism more likely. It was that latter part that was tested and found to be correct in cooperatively breeding birds and mammals (Cornwallis; Lukas). Your phrasing is so general that this key point is glossed over, which muddles the water. In cooperative breeders monogamy is a statistical predictor; for the evolution of true workers it becomes a necessary condition. These are very different things (see general comments above).

Line 90-91. 'enhance, retard, or have no effect ...' as 'overall conclusion'. Of what, of models that did not make their assumptions explicit? I am not aware of any empirical data set that has shown that the monogamy hypothesis was incorrect if you consider (as you should) that it never intended to replace Hamilton's rule in vertebrate cooperative breeders and social insects where altruism is facultative (condition dependent).

Line 92-95. This is true, but it seems to be just restating Hamilton's rule. No monogamy paper every claimed to replace that logic.

Line 96-105. My compliments for the constructive language in this paragraph. This is what the tone should be throughout this essay. You might want to add some of the 2010-2011 papers by Van Veelen as good examples of acrimonious inclusive fitness critique.

Line 106-113. It is unclear to me why the tone of this paragraph then becomes unbalanced again. I think 'prosecutorial' is once more a type of language to be avoided. All these papers did was reason from first GENERAL principles against a paper that was broadly conceived as a very unreasonable attack on massive progress in general understanding of social evolution based on inclusive fitness theory (see my general intro comments). It was NTW who started ignoring all the evidence (which you acknowledge in the previous paragraph). Objections were not 'disingenuous' (again wording that should be removed). All we can say is that there is disagreement about whether single-locus modelling with strong discontinuous effects on assumed phenotypes is useful or not (see my general comment above). Whether the mathematics was correct is besides the point. The equations were correct in all papers as far as I know, else criticism would have been much stronger. The point is that when assumptions are wrong, the outcome will make no sense. In such cases results are not inconvenient (another value laden term to be reconsidered), but irrelevant for what happens in natural populations. Please rephrase this paragraph to achieve balanced coverage as you do in the preceding paragraph.

Line 112. ‘... exceptions accumulate’ I am not aware of such accumulation. Do you mean the same four papers addressed above (refs 17-20)? You would have to give new references to justify ‘accumulate’.

Line 113. ‘uncritical’? To my knowledge almost every test of inclusive fitness theory has been an independent challenge of the general applicability of first-principle inclusive fitness theory. Critical attitudes in the Popperian sense have thus abounded and the theory has survived these refutation assaults at least in a qualitative sense. You may find that having more quantitative matches would be desirable, and I would agree. However, even the qualitative matches have been inspiring enough to generate new avenues of derived inclusive fitness theory, as for example David Haig’s genomic imprinting theory. By the way, also Dave Queller and Steve Frank have always argued that both types of models are useful and complementary.

Line 114. Remove the value-laden word like ‘presciently’

Line. 118-121. NTW did not pioneer simulation modelling and simulation modelling is something else than formal mathematical modelling. Formal mathematics provides first principle theory such as Hamilton’s rule and its broadened version using the Price equation. Simulation models address specific cases where multiple restrictive assumptions need to be made, and where complexity exceeds what can be tracked analytically. As I tried to explain in my general comments above, simulation approaches can be useful but only if the assumptions about what phenotypes do are reasonable, and they should never claim to demonstrate something truly general. Overselling in that way was typical for the NTW paper and because their assumptions were often wrong or irrelevant, it gained no followers. I do not understand why the author is so adamant to defend this paper as some kind of a ‘flagship case’. It detracts from the merits of his own models, which are more realistic and where he is willing to be specific about assumptions and about matches with general theory.

Line 124-125. ‘could consider’. Yes, but again the power issue is fundamentally different for lineages where colonies have facultative altruism versus those that have obligate altruism. You should make explicit that your point is about the former, not the latter type of social systems, and cite the well-known review by Beekman and Ratnieks to build any new argument on that synthesis. In fact it turns out that not only this first point, but also the following points 2-6 are about social systems with facultative altruism (see my last general comment above).

Line 136. ‘can easily select’ As far as I am aware there is no evidence for this to be ‘easy’ and to my knowledge there are no empirical data showing that selection for genetic diversity has produced consistent group adaptations in cooperative breeders. Reference 41 reports patterns across empirical studies but without addressing whether drifting ‘workers’ are potential reproducers or not. The only exceptions of unambiguous group adaptations owing to higher (chimeric) genetic diversity within groups appear to have happened after the point of no return to superorganismality had been passed. I.e., honeybee and leaf-cutting ant colonies deal better with disease when they have more patrines and costly obligate multiple queen mating is therefore a functional adaptation. Whether drifting workers can be considered to be a group adaptation remains to be proven and seems unlikely from a general theoretical perspective – these phenomena are merely effects of something else that could easily not have a functional adaptive significance (see Williams, 1966, for an important discussion about function and effect). Referring to ‘indirect reciprocity’ is unhelpful if you do not add a few lines of explanation of how that concept connects to what this essay is about.

Line 140-141. The monogamy hypothesis has always emphasized life-time (obligate)

unmatedness, not obligate sterility, which requires diploidy to be an automatic consequence of unmatedness.

Line 144-146. This is a hand-waving statement. The monogamy hypothesis predicts (both in the early reviews and more explicitly in the later ones, which you should cite if you aim to be balanced) that such reversals should not have happened. That prediction has been consistent with all available evidence so far (see Boomsma & Gawne 2018 and Smith et al. 2018 for updates), so a statement like this is misleading unless you can cite empirical papers that have provided evidence to the contrary. A more productive approach would be to make sure you acknowledge that the 'monogamy window' hypothesis for major transitions to caste societies is something else than facilitating monogamy in cooperative breeders (see my general comments) – and focus on the latter.

Line 147. The monogamy (window) hypothesis is about how facultative altruism can be replaced by obligate altruism, not about cooperation.

Line 148-150. Yes, it should work both ways, but only if relatedness is a true variable as in cooperatively breeding vertebrates, *Polistes* wasps, halictid bees etc. Once more, the core idea of the monogamy hypothesis is that strict full-sibling relatedness makes the relatedness terms cancel. That was apparently a necessary condition for major irreversible transitions to higher organismal complexity (both colonial superorganismality and obligate multicellular organismality), but not a sufficient one (the Leggett et al. paper – your ref 7 – also makes this point). Only long-term consistent $b > c$ benefits will forge major irreversible transitions after the necessary condition is fulfilled and that is a tall order. This entire point 3 is fine if you make explicit it is about cooperative breeders only (just like the other 5 points). In such 'societies' monogamy is never strict. References 4 and 8 are perfectly up front about that. Reference 3 was not, but you cannot hold that against the monogamy hypothesis. It is easy to plot the major irreversible transitions on the Hughes et al (2008) phylogeny (the base of the corbiculate bees, the base of the higher termites, the base of the vespine wasps). What you say here applies basal to those transitions, not beyond them – the later monogamy reviews have been very explicit about that, but that logic is also in the 2007 and 2009 versions. Irreversibility of transitions to coloniality where all individuals belong to a complementary caste for life (Wheeler superorganismality) is not just an assumption, it is consistent with all comparative data both for superorganismality and obligate multicellularity (cf Smith et al. *Nature E&E* 2018, Fisher et al. *Current Biology* 2013, Boomsma & Gawne, 2018).

Line 151-156. Yes, *Polistes* colonies and human groups are societies, albeit very different ones because we have cumulative culture driving our social evolution. So again this is a 'society' argument, but it has no bearing on the strict type of monogamy window hypothesized to represent a necessary condition for the emergence of obligate altruism in superorganisms (which no longer are societies of totipotent individuals). Once more, it would be useful if you somehow made these distinctions explicit when you intend this essay to be a way forward.

Line 158. 'sometimes'? Sibling competition within groups is generally and principally important as Hamilton and May (1977), Steve Frank (*JEB* 2012), Stu West & Ido Pen, and Leggett et al. (your ref 7) have amply documented. The group benefits you have in mind here are just the Hamiltonian $b > c$ values. Also your point 5 is about family groups before permanent castes evolve.

Line 162. And so is point 6. The Riehl work is very interesting, but not as a challenge of the monogamy hypothesis for altruism driven by indirect fitness benefits, because two key conditions do not apply: 1. 'helpers' gain no inclusive fitness – they just hang around to receive

prolonged parental investment. 2. These greater anis are not family groups where altruism could have evolved.

Line 168-172. The final quote is nice, but should in my opinion refer to the need to test both the assumptions and the predictions of any model used to better understand social evolution. It has no specific bearing on ecology and group-level benefits, which are fully covered by Hamilton's rule as the series of Steve Frank papers in JEB (2011-2013) amply document. Does it have bearings on genetics? Yes, quite possibly so, but remember Hamilton's rule is about genetics as well. If you want to make the genetics more explicit that can be useful, but it will always imply making restrictive assumptions that the Price equation (general inclusive fitness) approach does not need to make. If you keep adding special assumptions in order to gain 'realism', you may end up with a model that is very precise about a scenario that may never apply in nature. See my general comments and the Queller quote given above on how generality and precision of models trade-off. Both approaches are valuable when assumptions about individual and colony phenotypes are reasonable – and then they are complementary ways towards understanding. Models with inappropriate or unspecified assumptions are not useful. I think that should be a major condition for making your 'verify' by modelling credible, but the final arbiter of course remains what the empirical data show.

Koos Boomsma, Department of Biology, University of Copenhagen, Denmark

<https://orcid.org/0000-0002-3598->

Author's Response to Decision Letter for (RSOS-180913.R1)

See Appendix B.

Decision letter (RSOS-180913.R2)

19-Dec-2018

Dear Dr Nonacs,

I am pleased to inform you that your manuscript entitled "Hamilton's rule is essential but insufficient for understanding monogamy's role in social evolution" is now accepted for publication in Royal Society Open Science.

on behalf of Dr Alexander Ophir (Associate Editor) and Professor Kevin Padian (Subject Editor)
openscience@royalsociety.org

Associate Editor Comments to Author (Dr Alexander Ophir):
Dear Dr. Nonacs,

Thank you for your responses to the last round of reviewer comments. At this point I feel that you have done as good a job as you can to represent your point of view while also trying to accept and address the constructive feedback that you have been given. It is becoming evident that you and Dr. Boomsma (self-identified reviewer 2) have strong disagreements for many things, but with a few places of agreement.

I commend you both for trying to look past your steadfast disagreements and trying to produce a manuscript that offers an alternative that is, as far as I can tell, valid but unpopular. For this reason, I think your paper will be of great value to the literature. For the critics that give it a fair read, your paper can serve as a counter point to generate experiments and models that might help strengthen their point of view and address your critique. For those that are more agnostic or supportive of your point of view, this can serve as a call to arms to think about this problem differently and also to generate experiments and models that will move the field forward. In all of these cases, this will be a win for the community. There may be some critics that read and represent your paper unfairly. It sounds like this will (unfortunately) be nothing new to you. The best we can hope for in science is that we do our best to seek and find truth. I appreciate your efforts, and the efforts of the reviewers to this end.

Kind regards,
Alex Ophir

Appendix A

Reviewers' Comments to Author:

Reviewer: 1

Comments to the Author(s)

In this comment, Prof. Nonacs gives his view of the inclusive fitness controversy and related issues, partly through the lens of two recent papers: Olejarz et al (2015, eLife) and Davies and Gardner (2018, Royal Society Open Science).

Olejarz et al (2015) analysed a model of the evolution of voluntary worker non-reproduction in haplodiploid insect societies and found that monogamy sometimes promotes and sometimes inhibits the evolution of worker sterility. Davies and Gardner (2018) analysed the same model and found that the results of Olejarz et al. were due to restrictive assumptions concerning the genetics of worker sterility. When we (Davies and Gardner) relaxed these assumptions, we found that monogamy actually has a clear promoting effect on the evolution of worker sterility.

I have to say that I don't agree entirely with Prof. Nonacs' interpretation of my paper, but I do appreciate that he is trying to propose a way forward here and so I'll mainly restrict myself in this review to commenting on what I see as a few small matters of factual correctness. I don't think any of these problems are fundamental to what is being said in this comment piece more generally, but do need to be addressed.

Lines 82–84: "Davies and Gardner [24] expanded this analysis by allowing sterility alleles to be intermediate in their effect and beneficial to the queen in allowing her to gain more reproduction."

The first part of this sentence is right, but the second part doesn't seem right to me. Olejarz et al. made the assumption that the p_z function (proportion of male eggs produced by the queen) would be an increasing function of z . We adopted the same assumption. Therefore I don't think this can be described as a way in which we extended the analysis of Olejarz et al. To be clear, we did extend their analysis, but we did so by (1) allowing sterility alleles to be intermediate in their effect, (2) considering the long-term evolution of worker sterility rather than the short term invasibility of particular alleles, and (3) looking at a broader range of worker sterility scenarios.

Reply. Revised as suggested.

On a related note, further up in this paragraph, it is stated that Olejarz et al. assumed that "the production of [worker-derived] males competes with and possibly reduces the number of males that are queen-produced." This is technically right but I think phrased in a way that could be clearer. Olejarz et al. made the assumption that queen male production increases with worker sterility, so I feel like the word "possibly" isn't quite right here. Overall production depends on both r_z and p_z , so you *could* get a situation where worker sterility causes everyone's production to decrease (decreasing r_z), which *could* reduce the queen's male production overall (depending on p_z), but I think overall the way this is described here might be a little confusing to readers.

Reply. "possibly" was replaced with "might" to give it a more indefinite feel. Davies and Gardner do a good job of parsing Olejarz's assumptions. As the point I am getting at is that the differing conclusions can be seen more as a function of model assumptions rather than mathematical error, I don't need replicate the entirety of D&G's arguments. Readers can jump into those papers for themselves.

Lines 92–94: Davies and Gardner [24] respond to Olejarz et al [12] by declaring that "monogamy always promotes... worker sterility" without qualifying that this requires traits to be expressed additively..."

We spend quite a lot of the paper exploring alternative scenarios in which worker sterility is dominant or recessive, so I don't know where this is coming from—we absolutely do not require traits to be expressed additively.

Reply. This entire paragraph did not survive the revision!

Lines 124–126: "The degree to which parents or siblings can impose complete or almost complete sterility on their group mates will determine whether the Olejarz et al. [12] or Davies and Gardner [24] models are more appropriate."

I'm not sure I understand this—we make the same assumptions as Olejarz et al when it comes to who is controlling sterility, so I don't think whether parents or siblings can impose sterility on their group mates has anything to do with whether our analysis is more appropriate than Olejarz et al.'s.

Reply. I ended up combining this point with #1 and this has been deleted.

Finally, I wonder if the author would consider toning down the last paragraph just a little. I have always appreciated Prof. Nonacs' point of view, which has come to me through peer review a couple of times, and I appreciate that he makes his points in a measured way and does not seek to bury papers that he disagrees with. I also agree in general with the assessment in this paragraph that diversity of opinion can be fruitful and that excessive dogmatism is unhelpful. But I don't think I or my coauthor can fairly be described as people who consider "proving their opponents wrong" to be the most important thing, and it feels a bit like that is what is being said here. I worked on this paper during my PhD and I would never want to discourage anyone from wanting to study social evolution. In fact, I met with Carl Veller (one of the authors on Olejarz et al, who was a PhD student like me at the time) in person to ask him for feedback on the paper, and because I didn't want the authors—especially the less senior ones—to feel like there was any bad blood between us. So although I'm sure this wasn't Prof. Nonacs' intention, the final paragraph feels a little bit unfair to me.

Reply. What I originally wrote here reflects a lot of what I've heard people say in interviews, at meetings, and sometimes in confidential messages to me as an editor about the 'other' side. And I've found that both frustrating and counterproductive. However, my intention was for this comment to end on a more positive note and I see where the closing paragraph does not achieve that. It has been toned down!

Best wishes,

Nick Davies

Reviewer: 2

General reply. It surprises me that this reviewer is so focused on the 'monogamy hypothesis', which was specifically mentioned only once in the original version (and appropriately relative to a paper that was explicitly modeling it).

As a response to the same objection being raised multiple times below, I would question the reviewer's contention that somehow the initial evolution of facultative reductions in fecundity differs from the evolution of obligate sterility in the predicted effect of higher relatedness (e.g., as through monogamy). The basic Hamiltonian inclusive fitness relationship is the same: individuals evolutionarily sacrifice direct fitness to increase indirect fitness. Certainly, a number of papers have enthusiastically adopted the "hypothesis" in the context of the initial evolution of cooperative group living. Indeed, Boomsma et al's (2011) published response to Nowak et al. (NTW) directly claimed monogamy as the vital precursor to both the initial evolution of reproductive altruism and the later evolution of sterile castes. Granted, the rebuttal dropped the word "hypothesis", but it would seem inconceivable that the authors did not think the same dynamic was involved.

However, this reviewer's discomfort suggests that I missed my goal of showing the debate over the role of monogamy as the best exemplar of the controversy between those attacking inclusive fitness theory and those defending it. I have revised the manuscript in multiple ways that I believe better achieves my intentions. I also clearly delineate when I am considering the initial evolution of facultative sterility versus the obligate version.

Comments to the Author(s)

This manuscript evaluates the extent to which monogamy promotes altruistic cooperation. It is not completely clear to me why Peter Nonacs appears to feel that there are scientific reasons for considering possible alternatives to inclusive fitness theory for explaining patterns and processes of social evolution, which he does

in practice in spite of his closing sentence. Apart from the reference to Nowak et al (2010) nothing in his own work (which appears in several hypothesis-ideas towards the end of the manuscript) seems to justify that. I therefore believe that much of this essay sees controversy and disagreement where there in fact is none, provided one is clear about terms and definitions and the ways in which they were messed up when sociobiology became influential.

Reply. I honestly do not see how any objective reader of the literature since 2010 could conclude that no controversy or disagreement over inclusive fitness exists. To broaden the points I am trying to make, I cite a more complete set of dueling attacks and defenses.

In the four comments below I attempt to clarify this overall assessment. After that I provide specific comments with reference to line numbers:

1. Nowak et al. (2010) has nothing to do with the topic at hand. The reference to the Nowak et al. (2010) paper at the start of this manuscript is odd. That paper ignored the monogamy hypothesis (in spite of two Boomsma reviews of 2007 and 2009) having laid it out in considerable detail and the Science paper by Hughes et al. (2008) having shown it was supported by considerable empirical evidence. The Nowak et al. challenge of inclusive fitness theory was thoroughly refuted by a series of 5 rebuttals of which only one (Boomsma et al., 2011) explicitly addressed monogamy – the total number of authors involved was ca. 150 (not 134 as Nonacs states). The agenda of the Nowak et al. paper failed, because a scientific challenge of a major paradigm needs to achieve two things: 1. It needs to allow easy understanding of all insights already validated under the previous paradigm (in this case the 16 subfields listed with triple ‘yes’ in Table 1 of the Abbot et al. rebuttal that summarized the overwhelming empirical and modelling confirmation of inclusive fitness theory), and 2. Such a challenge needs to add a series of new testable predictions that were not valid questions under the existing paradigm, and that would have sufficient initial support to open up entire new research fields. This is what relativity theory (Einstein) and quantum theory (Bohr) did when they built on and enriched Newton mechanics. Nothing like that was achieved by Nowak et al. (2010). Their statement of ‘meagre’ support is simply wrong. Neither did their paper allow the recapture of any previous understanding or generate a single novel testable prediction. It therefore seems inappropriate to start the introduction of the present manuscript with Nowak et al. (2010). It would also be nice if Nonacs could actually mention more explicitly that the monogamy hypothesis has been very well supported by empirical data, not unexpectedly as it is in fact just an extension of Hamilton’s rule and the inclusive fitness theory behind.

Reply. NTW is clearly relevant to this discussion as it is the initiating event that spawned controversy and led to the continuing debate. True NTW did not specifically go after monogamy, but Olejarz et al is a direct descendent of the arguments presented there and, of course, Nowak is also a co-author. In the revised version, I make this overall connection clearer.

I believe 134 was the number of authors that signed the Abbot et al. paper (or was it 137?). While obviously I think NTW had something very useful to say, I also think it was a polemic that was excessive in several ways. I would note that I was one of the 150 respondents, where I published a (single authored!) complaint about their ‘example’ of the haplodiploid hypothesis as a failure of inclusive fitness. This idea had actually been rejected a number of years earlier, in part, by inclusive fitness type models!

2. Nonacs’ formulation of the central prediction of the monogamy hypothesis is incorrect. Nonacs appears to misrepresent what the monogamy hypothesis actually predicts. He is not the only one, because the Hughes et al. (Science 2008) paper testing the empirical evidence as provided by the then available comparative data is actually not very precise in how it phrased the idea being tested. If Nonacs is interested in evaluating the status of the monogamy hypothesis, he should therefore go back to its exact formulation by Boomsma, both in the first review cited by Hughes et al. (Current Biology 2007) and reread the precise phrasing in the review paper that he cites (Boomsma 2009). That should reveal that the monogamy hypothesis was ONLY meant to explain the evolution of life-time physically differentiated castes, not the evolution of all altruistic cooperation as the title of the present Nonacs manuscript suggests. It was stated explicitly that this concerned the evolution of superorganismality sensu Wheeler, which Boomsma coined ‘obligate eusociality’ struggling with the ‘eusociality definition’ that Wilsonian sociobiology had left the field with. It has become increasingly clear (and has been

expressed in increasingly explicit language by Boomsma (2009, 2013), Boomsma, Huszár & Pedersen (2014), and Boomsma & Gawne (2018) that sociobiology terminology is a smokescreen because its 'eusociality' does not discriminate between first helpers at the nest and life-time differentiated castes. The monogamy hypothesis never claimed to explain all aspects of what determined first helpers at the nest. It is about predicting the emergence of obligate altruism, not of facultative (condition-dependent) forms of altruism (facultative eusociality in the Boomsma reviews). Nonacs has made this interpretational mistake in his earlier modelling papers, which he now admits (e.g. nest inheritance), but his present text continues to repeat some of the same misapprehensions. His arguments thus acquire several strawman characteristics and they are already apparent in the title of the manuscript (the monogamy hypothesis is not about cooperation because reproductive altruism \neq cooperation).

Reply. In neither the original nor this revision am I particularly concerned with Boomsma's narrowly defined "monogamy hypothesis". Instead, I focus on the broader effect of relatedness across several aspects of social evolution. This is clearer in the revision.

Not everyone agrees with Boomsma on how to define eusociality. His definition, like every other one, ends up with problematic exceptions and inclusions. I prefer to look at process and not get distracted by human-imposed categorizations and definitions.

3. Assumptions should be explicit and biologically realistic for models to be meaningful. It is of crucial importance for evaluating the assumptions that go into any model meant to describe the evolutionary origin of different types of altruistic phenotypes. The previous Nonacs models have always addressed the origin of helpers at the nest, not the origin of obligatory altruistic castes, i.e. the irreversible transition to Wheeler-superorganismality which implies that ALL colony members belong to a single adult caste phenotype for life. The monogamy hypothesis never intended to challenge the recurrent origins of facultative (condition-dependent) altruism through the simple logic of Hamilton's rule ($rb > c$).

Reply. Not true. Nonacs (2014) specifically models this proposed evolutionary scenario.

If all three variables vary freely there can be situations where lower than full-sib relatedness still allowed some offspring to become helpers at the nest, for example because they were poorly endowed as adult phenotypes (maternal manipulation could play a role here, but note that mothers also need to weigh how they get most genes passed on to grand-offspring by making offspring disperse rather than stay). However, what the life-time monogamy hypothesis states is that it cannot be expected that this will ever lead to the evolution of permanently unmated workers (true workers as termite researchers aptly call them; neuters as Darwin and Wheeler conceptualized them). The monogamy hypothesis is about the evolution of obligate (unconditional) altruism, a distinction that is not made in the Hughes et al (2008) paper but that is explicit in the Boomsma reviews. Most examples that Nonacs highlights are about the origin of first helpers at the nest, and none is about helping behaviour going to fixation so that physically differentiated permanent castes appear. If Nonacs would make that distinction, most of the rationale for his critique would seem to disappear. The same rationale as outlined here also pervaded the recent Davies and Gardner paper, so the Nonacs critique appears to compare apples and oranges. A recent modelling paper by Quinones & Pen (Nature Communications, 2017) also shows clearly that strict monogamy is a necessary condition for the evolution of obligate, not facultative reproductive altruism. The Nonacs (2011, 2014) papers thus appear to have their assumptions wrong for them to be relevant as tests of the monogamy hypotheses for permanent castes – they address the evolution of first helpers.

Reply: Again, I believe that the initial evolution of helping behavior and the evolution of sterile worker castes is an apples to apples comparison. Quinones&Pen is interesting and illuminating in its own way (moreover this paper is also about the origination of facultative behavior and not the evolution of obligate castes!). However, nowhere in their model do they allow offspring to inherit nests. Hence, their results mirror those of Fromhage&Kokko. I suspect if you include inheritance, then you'd get something like Nonacs (2011).

4. The proposed 'alternative' hypotheses towards the end of the manuscript. Some of these appear to be interesting research agendas, but others seem rather far-fetched and lack more than hand-waving justification. More importantly, however, very few if any of them are directly relevant for testing the monogamy hypothesis in

its original meaning.

I hope the following line-by-line comments will clarify these general points.

Line 26: 'eusociality' here lumps societies and superorganisms, which Boomsma & Gawne show are fundamentally different social systems, going back to Darwin, Weismann, Wheeler, Fisher, Williams and others all the way through the 1960s, until sociobiology started to water down and muddle these definitions. The Boomsma reviews (particularly 2013) also make this point, emphasizing that Crespi & Yanega (1995) and Beekman et al (2006) also criticized the 'eusociality continuum' ideas emanating from the watered down definitions by E.O. Wilson in the 1970s.

Reply. The definition of eusociality is a philosophical point and the reviewer will find any number of people disagreeing with Boomsma. Fortunately mathematical models are blissfully insensitive to what we call things. Thus, Hamilton's rule is widely applicable across taxonomic groups and different manifestations of cooperation. That is its strength and value. But, I did drop "eu" from this sentence.

Line 29: Refs 9+10+12 had their modelling assumptions wrong for the monogamy hypothesis as originally formulated by Boomsma, which ref. 12 partly rectifies and the recent Davies&Gardner paper makes more explicit. You also admit that later on, but without stating the explicit reasons. There is also a semantic issue here. Empirical data can never be 'put to mathematical tests', not even in physics. Mathematical models can be used to derive predictions from axiomatic assumptions that can then be empirically tested, not the other way around. Being precise about this distinction is crucial particularly in evolutionary biology, where the main function of models is to make sense of empirically observed variation in social behaviours and life-histories – anything beyond that is illusionary.

Reply. To be clear, the tests are whether or not a prediction from Hamilton's inequality holds when placed into an evolving population model. It often does not (Lines 40-41).

Line 30: The monogamy hypothesis is not about the evolution of cooperation (for mutual benefits) but about the evolution of obligate reproductive altruism. What Hughes et al. (2008) and Cornwallis et al. (2010) tested was whether low parental promiscuity favors facultative reproductive reproductive altruism (with for social insects obligate reproductive altruism as a putative later extension). They never tested the hypothesis in its most stringent version originally proposed.

Reply. True, but then those papers cannot be cited as support for the monogamy hypothesis, as they often are - including in this review!

Line 32-33: When you conceptualize the evolution of permanent true worker castes as an evolutionary point of no return (i.e. a major evolutionary transition), as all Boomsma reviews since 2007 have done, the later transitions to multiple queen mating and polygyny (the latter primarily in ants – see Boomsma, Huszár & Pedersen, Animal Behaviour 2014) are not relevant for the argument, because by then true worker castes could no longer go back to facultative (condition-dependent) altruism. Also here 'cooperative behaviour' is an inaccurate description of what the monogamy hypothesis is about (same in Line 35).

Reply. That a sterile worker caste cannot evolutionarily revert to being fertile is an assumption and not a proven fact. To my knowledge, it has never been explicitly demonstrated. Given that sterile workers contain all the needed functional genes to be reproductively capable, and that much of social evolution involves changes in regulatory genes, it seems eminently possible for evolutionary reversions to at least arise. Alternative evolutionary pathways in which the worker or queen castes are simply lost (as in parasite or gamergate species) are common. Again, to my knowledge none of these are correlated with reduced relatedness. None of Boomsma's papers have ever convincingly argued for why relatedness (e.g., through monogamy) appears to be a unidirectional force working only towards increased division of reproduction.

I now include this as one area that might be fruitful to look into (Lines 139-150)

Line 35-40: Once more, the monogamy hypothesis is not about these cases. If daughters stay to help even

though they could have dispersed, there is no reason to question that they do that because the inequality specified by Hamilton's rule compels them to do so. Because these cases represent facultative altruism, it is expected that measured average direct and indirect fitnesses are in fact equal (representing a frequency dependent ESS equilibrium where both strategies do equally well). This is what the cited studies report and it is all consistent with inclusive fitness theory, even though the papers try to argue otherwise – very little if anything in these decision making processes can be shown to be maladaptive or challenging established inclusive fitness theory (line 41-42). But the main issue is that none of this challenges the monogamy hypothesis for its prediction of major evolutionary transitions to superorganismality as defined by Wheeler (Boomsma & Gawne, 2018).

Reply. The reviewer is completely wrong here. In each of the 4 papers cited there is no evidence for a frequency dependent ESS in which females that stay to help do equally well as dispersing females. 'Choosing' to become a helper is a maladaptive decision from the helper's perspective.

Line 44: The above implies there is no dissonance between theory and empirical results as implied here, provided one phrases the predictions and assumptions of the Boomsma hypothesis correctly, rather than forcing them into uninformative sociobiology jargon that Boomsma tried to avoid.

Reply. There certainly is dissonance across Fromhage & Kokko versus Nonacs in terms of each paper's conclusions! The point of the paragraph here is that their differences follow entirely from a difference in one assumption that each model makes. Thus, neither paper is wrong from a mathematical perspective. They are just modelling different worlds. My argument in this comment is that this same pattern of dueling models on some aspect of social evolution reflect more about differences in assumptions rather than that one method is flawed.

Line 50: Again, the argument is not about cooperation but about reproductive altruism.

Reply: OK, revised.

Line 56-58: How do you define 'a species' sociobiology'? It would seem preferable to me to avoid vague language like this. Once more all of this is about the expression of condition-dependent facultative altruism with or without simultaneous direct fitness gains (captured by mutualistic cooperation) – nothing of what is here appears to challenge the monogamy hypothesis for the evolution of obligate reproductive altruism.

Reply: Replaced sociobiology with ecology. However, the argument has nothing to do with condition-dependent facultative altruism. Instead, it is about the fitness consequences for a given behavior across differing social life histories.

Line 59-63: Also this point seems misplaced as criticism of the monogamy hypothesis, because it seems to represent a 'parasocial' arrangement, which the Boomsma reviews explicitly excluded as ancestral state for the evolution of superorganismality in the Wheeler sense.

Reply. In rereading the paper, I realized the points being made in this paragraph were out of place relative to the flow of the rest of the manuscript; causing confusion. It has been deleted.

Line 63-67: This makes sense to me, but it is perhaps premature to highlight here based on just a submitted paper, particularly because all of this seems consistent with the Hamiltonian paradigm that ecology (the b/c ratio in Hamilton's rule) is as decisive as the relatedness terms when all three vary at the same time, as is always the case when evaluating cases of condition-dependent expression of facultative reproductive altruism. All the monogamy hypothesis did was claim that maximal and invariable relatedness was a necessary condition for making major evolutionary transitions to obligate reproductive altruism with permanently unmated workers – so for this kind of transitions relatedness comes first, cancels out of Hamilton's rule, so that $b/c > 1$ comes to represent the sufficient condition for actually making such transitions. A recent paper (Smith, Kent, Boomsma, Stow, Nature Ecology and Evolution 2018) shows that this proof of concept

works fine to explain the evolution of life-time unmated workers in an ambrosia beetle – the commentary by Nick Davies summarizes that very well.

Reply. Part of the deleted paragraph.

Line 67-69: I think this point has been made earlier in a Keller and Reeve model – all fine. It just has no direct bearing on the assumptions and predictions of the Boomsma monogamy hypothesis.

Reply. Same deleted paragraph.

Line 70: ‘Recently the monogamy debate has extended ...’ ?? If you carefully read the Boomsma reviews all the way back to 2007 you will see that this was the core of the argument right from the start – even though the language was made more explicit in later reviews. Ref 10 confuses the evolution of first helpers at the nest with the point of no return to obligate reproductive altruism, which are very different things. Even though some others have also confused these issues in previous papers, you cannot hold the logic presented by Boomsma responsible for that confusion.

Reply. I absolutely disagree – Nonacs 2014 is not about 1st helpers. Note that the title of the paper is: “Resolving the evolution of sterile worker castes: a window on the advantages and disadvantages of monogamy”. It is a direct response to Boomsma’s 2013 claim that prior models (such as Nonacs 2011) were examining the wrong evolutionary transition. Thus, the model starts with a cooperative group with reproductively capable workers. It then examines under what conditions can worker sterility spread. The findings are that sometimes monogamy is favorable, sometimes it is irrelevant, and sometimes it retards the spread of sterility.

With the deletion of the distracting paragraph immediately before this one in the original version, there is now a clear delineation between models testing the origin of facultative sterility and those testing for obligate sterility (Lines 56-87).

Line 75-76: No, of course not. But maternal manipulation is only a (partial) alternative explanation to voluntary reproductive altruism in social systems where some nests retain helpers and other nests do not. All those dynamics are covered by the classic dynamics of Hamilton’s rule and have no bearing on the monogamy hypothesis as formulated for the evolution of life-time unmated nursing castes.

Reply. Maternal manipulation is an alternative explanation to voluntary sterility (and not partial at all – see Kapheim et al. 2015, #19 in the References). Monogamy is irrelevant with manipulation; hence the importance of sibling relatedness depends entirely on the degree to which parents can and do manipulate offspring into subservience!

Line 76-87: For all I know, Davies & Gardner showed that the Olejarz et al. model was unsatisfactory because it worked with caricature single locus variation with extreme phenotypic effects. Ever since Alan Grafen wrote his well-known phenotypic gambit paper in the 1980s the field has been clear about complex social traits being largely independent of such single locus extreme effects. Making models of that kind (as Olejarz, Nonacs, and some others like to do), therefore has possible bearing on the first (facultative) expression of reproductive altruism, but not on the evolution of permanently sterile castes, which requires significant rewiring of complex gene-regulation networks depending on many genes with small effects. That is why social traits are best considered to have quantitative genetic variation with overall heritabilities that determine how such traits respond to selection, but without significant single-gene effects. I would think this should also apply to many cases of the expression of facultative helping when governed by the type of phenotypic plasticity proposed by Mary Jane West Eberhard. Also here it is hard to conceptualize how complex behavioural reaction norms are dependent on single locus variation with extreme phenotypic effects.

Reply: As a criticism of Olejarz & Nonacs, this equally applies to Boomsma’s work, which by framing the monogamy hypothesis as purely an inclusive fitness question neglects the same complexity. Boomsma just assumes the complexity will align with his hypothesis. So I am not sure what the reviewer’s point here is. I would argue that Davis&Gardner, Olejarz et al & Nonacs are at least a step mathematically closer to the actual evolutionary genomics. I don’t expect that any of these papers have given the final answer, but they have gotten closer to it.

Line 98-100: I disagree with this blanket statement, because it suggests that all arguments have equal merits across the board. The Boomsma monogamy hypothesis has met with unambiguous support for the predictions that it actually made (major transitions to Wheeler superorganismality and to obligate multicellularity cf. Fisher, Cornwallis & West, *Current Biology* 2013). Other models, among them some by Nonacs, have shown that things are more complex for facultative reproductive altruism. Under the assumptions stated, these models are likely correct, but they are not at variance with the monogamy hypothesis as originally framed. They shed light on cases of haplodiploid cooperative breeding and complement more general insights in (usually diploid) cooperative breeding in other arthropods and vertebrates. When taking that broader perspective, haplodiploidy is actually quite an idiosyncratic issue, because it implies that unmatedness does not necessarily imply sterility as it does in diploids. I think also that should be taken into account when evaluating the merits of a loser version of the monogamy hypothesis that has now been tested and found to be essentially correct in many diploid cooperative breeders (alpaeid shrimp, cooperatively breeding birds, cooperatively breeding mammals, etc – see recent Boomsma reviews for references).

Reply. It is true that Fisher et al. find that most transitions to multicellularity they surveyed were from clonal ancestors, but a significant fraction were not. It is a matter of personal preference as to whether or not one considers that to be “unambiguous” support (also, one might argue that to be consistent with the objection raised multiply in this review – it is not a test of the monogamy hypothesis of facultative sterility evolving into obligate sterility). However, one must also consider that multicellularity requires single cells to be in close proximity – a constraint often easier to overcome within a clone than with non-clones. Anyway the evolution of multicellularity is a different paper to be considered on a different day.

Line 104-142: This list is likely to be in need of qualifications as current relevance seems doubtful. 1. Bourke & Franks (1995) argue that maternal manipulation is essentially a Hamiltonian concept, as was agreed upon by Richard Alexander, so any discussion of the validity of this idea should be framed in that context. Also manipulating mothers are under selection to maximize their inclusive fitness, so manipulating provisioning of daughters in their larval stages so they stay should not give mothers fewer grandchildren than provisioning them optimally so they can disperse and breed.

Reply. Kapheim et al 2015, found that mothers are maximizing their fitness at the expense of daughters that stay as helpers. The clear and obvious point is that if helping is a manipulated result, then the question of monogamy or not, is moot.

2. So far ‘social heterosis’ has never been shown to play a role empirically for facultative reproductive altruism; it has for obligate altruism in honeybees and leaf-cutting ants with obligate multiple queen mating. It seems crucial to acknowledge that distinction.

Reply: I would point out that clonal multicellularity has evolved perhaps around a dozen times in the history of the earth. Interspecific facultative and obligate mutualism has evolved independently perhaps millions of times. This is social heterosis. Almost all multicellular ‘species’ are actually a close association of themselves and their microbiomes. Given we are all “holobionts”, social heterosis may well be involved. Phylogenetic analyses of cooperative breeding in the hymenoptera and birds find that the likely ancestors were monogamous. So that is two evolutionary occurrences. Within those phylogenies, multiple mating and lowered relatedness has evolved over 40 times independently – with no noticeable loss in cooperation. A 20-fold difference in evolutionary changes perhaps due to social heterosis. Maybe it is about time we actually looked for the positive effects of genetic diversity and low relatedness in intraspecific facultative cooperation?

3. I do not think it is appropriate to capture humans and paper wasps under the same heading – overall none of these two systems seem relevant for the monogamy hypothesis as originally framed.

Reply. I would prefer the readers to decide for themselves the value of this suggestion.

4. This seems misguided because the monogamy hypothesis is about obligate reproductive altruism (and in fact obligate multicellular soma as well) evolving ‘voluntarily’. That is why maximal (invariable) relatedness is key because it makes the r -terms cancel out of Hamilton’s rule. That makes the power issue brought up here moot. The Davies & Gardner model acknowledges that, any model coming out of Nowak’s group does not.

Reply. I have merged the idea of control here with point #1, which is similar.

5. As far as ‘groups’ are parasocial aggregations, this argument is irrelevant for the monogamy hypothesis which requires colony founding by a life-time monogamous pair.

Reply. However, it is relevant whether grouping with close kin or not is fitness enhancing.

6. Also this point seems hardly relevant, because outbreeding is normally secured in social systems. The few cases in which societies have offspring hanging around that do not obtain indirect fitness are irrelevant for the monogamy hypothesis. As Michael Griesser argued (recent PLOS Biology paper) there are forms of family life that essentially represent extended parental care. They will imply interesting social dynamics, but there is no relevance to the monogamy hypothesis when offspring do not help to obtain indirect fitness benefits.

Reply. It may be important as regards monogamy when one sex can gain direct fitness in the natal nest and the other cannot. Short answer – I don't know, but it would be interesting to consider.

7+8. Also these topics do not seem relevant for the monogamy hypothesis. They are of interest, but have been of interest ever since inclusive fitness theory was proposed so are hardly novel; antagonistic interlocus genetics remains poorly understood even in sexual selection research – it seems hardly relevant as a pressing research agenda here.

Reply. #7 is deleted, #8 I realized is just another way of expressing #5 and they have been merged.

In sum, the title of the paper is too broad for the issues the paper actually evaluates. The monogamy hypothesis was never about cooperation for mutual direct fitness benefits, but only about the evolution of obligate reproductive altruism. Suggesting it was more by using 'cooperation' muddles the water. Many of these 1-8 topics are of interest, but just not in the context at hand.

Reply. Title has been changed to reflect more the overall thrust of this comment.

Line 145-149: All valuable scientific theory is dogmatic. How else would Einstein and Bohr have succeeded replacing Newton mechanics? G.C. Williams explicitly emphasized the need for biological theory to also be stringent, so its predictions are vulnerable to falsification – that is why he explicitly endorsed inclusive fitness theory in his 1966 book *Adaptation and Natural Selection*. Neither I nor any others who have developed or tested the monogamy hypothesis have ever claimed it will be relevant for any social system. However, the framing has been increasingly precise about what is predicted and what is not (see particularly Boomsma reviews from 2013, 2014 and 2018), and these exclude scenarios involving the expression of facultative reproductive altruism with less than maximal sibling relatedness or cooperation for mutual direct benefits.

Reply. In response to Davies, this paragraph has been much toned down.

Line 149-153: As far as I am aware we know of no examples where 'emergent group level benefits' have been shown to challenge inclusive fitness theory. In general, any invocation of 'emergence' is unhelpful for scientific progress. As J.B.S. Haldane wrote: 'Particularly hostile to true scientific progress are the extremer forms of emergence [S]cience is committed to the attempt to unify human experience by explaining the complex in terms of the simple - *The Causes of Evolution* (1932). I think Peter Nonacs adheres to this principle when he closes off writing "Trust in inclusive fitness, but verify". However, it would then help for maintaining transparency in the literature if he would refrain from sociobiology terminology that was never intended to accurately capture the assumptions and predictions of inclusive fitness theory, and from using strawman versions of extensions of inclusive fitness theory to line up arguments that do not address real issues.

Reply. Dropped "emergent"

Appendix B

Reviewer comments to Author:

Reviewer: 1

Comments to the Author(s)

As before, I am limiting myself to commenting upon issues of factual correctness, especially regarding Davies and Gardner (2018), rather than issues of interpretation or opinion. That's not to say that I think natural history is a matter of opinion. As the author states, people might disagree on how to define eusociality or any other word, but I don't agree that this is merely a "philosophical" issue because some definitions increase our understanding and others don't.

Because I'm an author of the piece that is being commented upon, I think this is the appropriate role to take here and although I don't agree with everything being said here, my feeling is that that debate is better held in the open.

Lines 26-29. I think the relatedness term in Hamilton's rule is incorrectly defined here, because it seems like c is the number of foregone offspring for the actor, and b is the number of additional offspring for the recipient, so then r should be the relatedness between the actor and the recipient, not the relatedness between the actor and the recipient's offspring. (Or alternatively, c needs to be multiplied by the relatedness to the actor's offspring.)

No, the equation in the paper is correct. The recipient in this case is the extra siblings reared. This is the benefit accrued to the colony. What the actor sacrifices (its cost) is the offspring it will not have. The confusion is that relatedness is often 'built into' the c term. Hence, the monogamy hypothesis posits that cooperation is favored when $r_1b > r_2c$. For one's own offspring, r_2 is always 0.5 (which is why it often is not explicitly stated), and for full sibs then the r values cancel and cooperation is favored when $b > c$. For half sibs, $b > 2c$ must hold for cooperation to be favored. This is the basis for any version of the "monogamy hypothesis". I made this explicit rather than implicit in the revision.

Line 40. This is just a comment. In my view, there are two (related) reasons why the cited models don't conform with Hamilton's rule ($rb > c$). One is that some of the studies assume that altruistic phenotypes are encoded by alleles of large effect. I hope it is becoming increasingly clear to theorists and practitioners that predictions made from models that assume quantitative genetics or which make equivalent assumptions such as additivity of allelic effects (as Hamilton did in deriving $rb > c$; also see previous comment on quantitative genetics from reviewer 2) cannot simply be transposed into a

model that assumes alleles of large effect. A lot of the disagreement over inclusive fitness really boils down to this issue. My own view is that the quantitative view is the more general one, since it is equivalent to the assumption that mutations of intermediate effect will eventually arise in a population. It also has empirical support for the reasons mentioned by reviewer 2. Finally, in cases where the predictions made by quantitative genetics and the predictions made by biallelic models diverge the most, that is precisely when there will be the strongest selection for an allele of intermediate effect to spread. So I really think there is good reason to believe that, over evolutionary time, populations will tend to conform to predictions made by quantitative-genetic models rather than models with alleles of fixed effect.

I have to say I have seen this objection raised more than once and wonder if it is more expectation than fact. I have sometimes for my own edification replicated my simulation models to just check this out, with multiple alleles that are additive in effect and with small or intermediate effects (plus with linkages, etc) – and gotten the same or very similar results. Beyond this, I don't see any universal justification for why a more quantitative view would always be expected to recapitulate Hamilton's rule – as it seems to be the assumption. No matter on which side of this argument you stand, I still think that the central point of my comment stands. There are better simulation methods to model the evolution of cooperation than the highly limiting inclusive fitness approaches.

The second reason is that Hamilton's rule—in the strict sense of $rb > c$ —is designed to analyse a specific social phenotype, in which an actor gives up some of their reproductive fitness in order to increase the reproductive fitness of a relative and there are *no other impacts to their actions*. One can't take $rb > c$ and apply it unthinkingly when there are other things going on, like male production by workers, nest inheritance, and so on. The way that Gardner, myself and others apply inclusive fitness theory is by first building a model which (hopefully) captures the relevant phenomenon, then interpreting the results in terms of inclusive fitness, not by starting with $rb > c$ and trying to fit it to every social evolutionary problem that involves altruism. For example, Davies and Gardner (2018) don't end up with $rb > c$, but a more complex condition.

I think this is entirely true, but often ignores the point of view problem. For example, in a social insect colony, there is the queen, non-reproductive workers and potentially reproductive workers. All have different versions of $rb > c$. Not everyone's fitness can be maximized simultaneously. To predict the outcome, one must therefore know who holds the power. Monogamy matters most if it is the potentially reproductive worker and not at all if it is the

mother. This is what Nonacs (2014) shows and my greatest problem with Olejarz et al and Davies & Gardner. Both look only one point of view. An interesting extension would be to what the outcome would be if power was shared across all 3 actors. I don't see how this question could ever be answered by an inclusive fitness model, even though Hamilton's rule is the genesis of what makes it an interesting question in the first place.

Line 83. Is there a word missing before "evolution"?

Fixed

Line 127. halictid typo.

Fixed

Line 181. My middle initial is G, not D.

My apologies!

Best wishes

Nick Davies

Reviewer: 2

Comments to the Author(s)

I appreciate the efforts by the author to clarify a number of points that I raised in my first review, which has given a better balance overall. However, my comments below will emphasize that further revision would be desirable if this paper is to serve the authors' purpose of pointing towards a way forward.

General comments:

- It remains puzzling to me that the author seems to think he can discuss the merits of monogamy as a key explanatory variable without precisely referring to where the idea comes from and how it developed. Before Boomsma (2007) the concept simply did not exist, except as a few throwaway sentences by Hamilton, Dawkins and Cronin (the introduction of that first review has the citations). If you choose a title like the one you have now, and aim to make a contribution that could ease a perceived controversy, you will have to relate to the concept as it was developed between 2007 and 2018 and at least briefly mention the consistent empirical evidence for it. This mentioning should differentiate between the strict monogamy (window) hypothesis for major transitions to permanent castes, and monogamy as reduced promiscuity serving as statistical predictor of the expression of

context-dependent (facultative) altruism. These two aspects are really different issues and not acknowledging that difference muddles the water. The central role of monogamy in determining social insect life histories and reproduction was first brought to high prominence by the Trivers & Hare 1976 paper, and then by my papers (among others) in the late 80's. In what is a short comment, there is not really the space to devote to the full history of how high relatedness through monogamy has been proposed to have evolutionary effects for many aspects of the sociobiology of Hymenoptera (consider that the Boomsma & Gawne paper that delves into the terminological history is 27 pages long!). I think it is clear enough that many believe monogamy to be quite important, while notable others have called the central role of relatedness into question.

More seriously, as in my previous response I absolutely do not agree with Boomsma's premise that the evolution of facultative and obligate sterility are evolutionarily different processes. The effects of monogamy, in the context of Hamilton's rule, are viewed as critical to both. Furthermore, I disagree that with the contention that obligate sterility in social insects is a major evolutionary transition (an MET). Whether or not it is, this an argument for a different paper and would only be a distraction in my short comment

- If you did (re)read the full spectrum of monogamy reviews, you would notice that monogamy was ONLY claimed to have been a universal necessary condition

Exactly!!! And this claim is what several models are specifically challenging!

(and NOT a sufficient condition) for the evolution of life-time unmated castes.

Exactly!!! And these other conditions, such as parental manipulation, would make the question of monogamy or not, irrelevant!

I really do not know how to make these points any clearer in my paper.

It has also been made clear from the start (and in increasingly precise language in later reviews) that in all societies of cooperative breeders (e.g. halictids, Polistes, lower termites, naked mole rats, snapping shrimp, meerkats, family living birds) the three parameters of Hamilton's rule always vary simultaneously (even when relatedness is high). This implies that the monogamy hypothesis had nothing fundamentally new to say about those social systems where multiple (often most) co-breeding individuals have independent agency (totipotency) – the hypothesis only claimed to parsimoniously explain the handful of convergent major transitions to obligate eusociality (Wheeler superorganismality; Boomsma & Gawne, 2018) where that independent agency has irreversibly disappeared. If you write (line 19) that 'No single topic better exemplifies the ongoing controversy than the

question of what role does monogamy play in the evolution of reproductive division of labor within cooperating family associations', you just have to be explicit that you address monogamy as it applies to societies of cooperative breeders, not the strict major transitions monogamy-window hypothesis.

The next sentence after the one quoted above, does exactly that. Again, the published papers examine both cooperative breeders and the supposed MET. It is fundamentally the same argument as to the value of Hamilton's rule and what types of models best elucidate the possible evolutionary progressions.

- With this in mind, I remain unsure whether I understand your title 'Hamilton's rule is essential but insufficient for understanding monogamy's role in social evolution'. I think we agree on the first part, but I do not understand the ending. The strict monogamy-window hypothesis is just a reduction of Hamilton's rule (the r terms cancel) and if monogamy is not 100% the standard version of Hamilton's rule continues to capture everything as a first-principle theory. So what is insufficient in your title supposed to mean?

Hamilton's rule, $rb - c > 0$ is the entire theoretical basis for Boomsma's monogamy hypothesis. Above he claims the hypothesis to be universally true. This is a logical fallacy along the lines of: "All people named John are men; therefore all men must be named John". Of course, all men could be named John, but it is not a certainty that they are. Thus, although Hamilton could be both essential and sufficient, a whole series of population genetic models (Nonacs 2010, 2014 and Fromhage and Kokko if properly evaluated) reveal that cooperation pays off more in a group where full sibs are helped, but across the entire population, more groups of half-sibs receive help. The two end up cancelling each other out and therefore the spread of cooperation occurs equally fast, independent of mating strategy. Hence Hamilton's rule is insufficient to easily or adequately capture this sort of probabilistic consequence.

We know that inclusive fitness is the sum of direct fitness and indirect fitness, so do you mean that direct fitness benefits are often essential as long as altruism remains facultative? But that is true by definition according to Hamilton's rule. Even if mothers would manipulate daughters against their interests (for which the evidence is weak, but getting stronger & stronger as another study – Snell & Rehan - has just been published while this comment was being reviewed), parental manipulation does not represent something alien to inclusive fitness theory.

Of course it is part of inclusive fitness theory, but the key point is that manipulation makes monogamy irrelevant.

Bourke and Franks (1995) have some pages of text explaining this. If this essay is to meant to objectively map out what we (dis)agree on and what is perhaps subject to debate, I think you should make clear that the monogamy (window) hypothesis predicted how major transitions to permanent castes have come about and then add that this may have confused readers to believe monogamy was an essential condition for all forms of social evolution. You could then proceed outlining why this is not the case and how/why Hamiltonian logic allows a variety of social traits to be maintained either for reasons of cooperation (no relatedness involved) or by facultative expression of altruism, or (perhaps) by offspring being manipulated into maximizing their mother's fitness. Anything else will be perceived as a broadside against the monogamy hypothesis as a whole that lacks the necessary nuance and will thus not achieve the balance you appear to aim for.

This comment is basically asking me to rewrite the paper to reflect Boomsma's views and perspectives on MET's and the role of Hamilton, when it is clear that I do not share them. The paper is in no way a blanket broadside on either monogamy or the effects varying levels of relatedness might have. Rather, I am pointing out that the issues are demonstrably more complicated and nuanced than Boomsma's rather simplistic black vs white view on NTW and the issues they and subsequent authors raise.

- Pitting the monogamy idea against NTW (2010), a paper that failed to make even a single testable prediction and that kept all its assumptions implicit, seems a tall order when the goal is to remain balanced. Modelling in ecology and evolution is ultimately about explaining variation in social systems of very different kinds from the same parsimonious set of first principles. Precise definitions are part of the transparent overall approach needed to achieve that – not merely a philosophical 'human imposed' issue. Evolutionary biology cannot be a serious science if one focuses on 'process' without having precise language to capture process, as the author argues in one of his responses. This contrast is particularly strange because NTW only modelled full sib colonies, so they implicitly worked with strict monogamy as your ref 15 pointed out. I still believe you do your mission a disservice by starting to argue from NTW (2010) in implicitly supportive language, because these authors were never interested in a proper discussion whereas you are. I agree that NTW (2010) was not an invitation to discuss. It was a "broadside" and unsubstantiated denunciation of a rather successful field of study. The follow-up papers from the group have also unfortunately continued in their over the top hyperbole, which concerns me greatly. Nevertheless, when one looks behind the rhetorical excesses, there are valid points being made. Hence, a discussion is warranted, even if the people who started it all refuse to do so graciously.

- Assumptions of models do matter crucially (as the author highlights in

line 63), but this arguments deserves to be generalized. In evolutionary biology models should both make their assumptions and predictions explicit, which is rarely consistently done for phenotypes in population genetic models.

Which is why Hamilton is so valuable – it gives you phenotypes to be modeled! I.e., the “essential” part of my title.

Furthermore, both assumptions and predictions should be testable. One cannot expect that tests of the predictions have been done when a model is novel, but scientific transparency requires that modellers tell their readers how their assumptions apply to specific groups of organisms and provide reasonable evidence that they are likely to be valid in a sufficient number of cases to make the model worth the trouble. The NTW paper is a case in point where essentially all crucial assumptions remained implicit and sometimes turned out nonsensical when digging them out (e.g. the idea that workers do not need to be considered as evolutionary agents). Another, already mentioned, implicit assumption in that paper was that they in fact only considered full sib families where relatedness was constant and non-variable. I find it hard to imagine that you endorse of practices like this and believe you have sufficient experience as a field naturalist to agree that matching model assumptions with empirical evidence is a point of key importance.

This is, of course, what I tried to do in Nonacs (2010,2014 & 2017). The results of the models were not always supportive of simplistic conclusions drawn from Hamilton’s rule such as the need for universal monogamy. No one has shown any of these papers to be wrong.

- The disagreement issue that you address in this essay is real, but not a discussion between parties holding equally strong decks of cards - less than a handful empirically inclined biologists have seen any merit in NTW (2010), about 30 times fewer than the ones who signed the 2011 rebuttals.

Truth in science is not a popularity contest where the majority vote is always right. Otherwise, we’d be standing on a 6000 year old, flat earth, at the center of the solar system.

This is why I think you do your mission a disservice by focusing on that particular paper in the approving sense that your wording suggests. The issue is really about types of models to be used and the assumptions that feed them. I think the author and I agree that the significance of inclusive fitness theory has been immense. What we also should agree upon is the ‘make assumptions explicit’ issue summarized above. What (only in part) separates us is that he appears to feel that specific population genetic models have the potential to overturn general first-principle theory such as Hamilton’s rule, where I feel they will only complement first principle theory.

Again, I don't think we have or likely will overturn Hamilton in the sense that $rb - c > 0$ will be shown to be universally or even often locally false (although never say never!). And I did and do state this (L 118-119; 168-169). I would use the word "elucidate" and not "complement" for the relationship of pop gen to Hamilton.

But at least here we have an issue that may well not be obvious to many biologists, so is worth clarifying because it never appears authors like Nowak, Van Veelen, Allen and E.O. Wilson have an interest in engaging in this discussion.

I do agree, but after several suggestions to "tone down" previous versions of this manuscript, it is perhaps wiser not to go there in print.

My overall impression is that it always turns out that when Nowak's group produces an explicitly genetic model, it is just a matter of time until someone else (e.g. Nick Davies and Andy Gardner is the most recent case) shows those 'overturning conclusions' are simply a consequence of making rather extreme assumptions about how genes affect phenotypes.

I would not necessarily call all the assumptions extreme. More like limiting, and then there usually is a rebuttal that critiques the critiques and points out their limitations and possibly flawed assumptions. Moreover, peer review keeps letting these papers, critiques and rebuttals be published, so they must be finding an accepting audience.

But that type of discussion is not new – Hamilton and later Dave Queller, Steve Frank, Stu West, Andy Gardner, James Marshall, Peter Taylor, Francois Rousset, etc have all made it clear that strong selection, drift, dominance, non-additivity, etc will make Hamilton's rule only approximately valid or sometimes make it fail. To see why, one should realize that as soon as one goes down the path of more specific 'realistic' models the extra assumptions needed (relative to the Price-equation inclusive fitness approach) make such models less general. As Dave Queller writes in the Introduction of his 2017 AmNat paper: 'Levins (1966) discussed how models have to trade off between generality, realism and precision. Kuhn (1977) similarly noted that theories face conflicts between accuracy, consistency, scope, simplicity', and fruitfulness. It is my clear impression that the author agrees with this but it remains very implicit.

- The non-novelty of the NTW critique on the Price equation approach has been aptly described by Steve Frank, a mathematical scholar on par with Nowak. If the author would care to check his series of seven papers in the Journal of Evolutionary Biology (2011-2013), he would find all the references to these early insights. Particularly Frank (2012) Natural selection. IV. The Price equation (JEB 25, 1002-1019) is illuminating, because the final sections address the so called 'controversy'. I quote: 'This quote from van Veelen et

al. (2012) demonstrates an interesting approach to scholarship. They first cite Frank as stating that dynamic insufficiency is a drawback of the Price equation. They then disagree with that point of view and present as their own interpretation an argument that is nearly identical in concept and phrasing to my own statement in the very paper that they cited as the foundation for their disagreement.'

I would say that the arguments over the Price equation still continue to this day, with quite notable and talented mathematicians on both sides. If there were an obvious answer and satisfactory to all – this would have been settled a long time ago.

Given this kind of arguing, does it then surprise the author of the present essay that little appreciation for the approach of NTW (2010) has accumulated? Another quote that the author may find interesting is from Frank's final paper in this series entitled Natural selection. VII. History and interpretation of kin selection theory (JEB 26, 1151-1184, 2013) which says: 'Nowak et al. (2010) say that one must make a specific model for a specific case, and then one gets the right answer. True, but the history of science shows unambiguously that one gains a lot by understanding the abstract causal principles that join different cases and different models within a common framework (Frank, 2012b).'

The upshot is, as Hamilton already knew, that explicit population genetic models are important, but they are (by definition) never general and generate few testable predictions.

I argue that pop gen models are in a sense, mathematical tests of inclusive fitness predictions. If the IF model 'fails' the test, the results provide insight as to why they failed. I think that my previous papers are exemplars of this. As for making testable predictions, yes they do. I would also point out, that testing the maximize fitness prediction from IF models is about the hardest publishing endeavor I ever faced as a researcher (see the Nonacs & Richard comment).

- In light of the above, I retain my opinion that, for this essay to be influential, you need to make explicit that the role of monogamy as a predictor is crucially different in social domains where altruism is facultative (i.e. cooperative breeders) and domains where altruism is obligate (all helpers remain unmated; hence they are true workers - what Darwin meant with 'neuters'). Boomsma & Gawne (2018) have the most recent and complete update of that logic. Making this distinction is important because it seems futile you challenge the evidence for the major evolutionary transition logic of the monogamy hypothesis given the consistent evidence (the Smith et al. paper on *Austroplatypus ambrosia* beetles in *Nature E&E* (2018) gives a complementary summary of what Boomsma & Gawne reviewed). You can then focus on the cooperative breeders where becoming a helper remains facultative, so that either social nests coexist with solitary nests (e.g.

Megalopta) or where reproductive success always requires colonial sociality but where relatedness is never high enough to create a 'monogamy window' (e.g. Polistes, naked mole rats, Seychelles warblers, meerkats). Focusing on that type of social systems is useful because your summary will then make clear that: 1. Monogamy may help, but not necessarily does help (e.g. Megalopta if maternal manipulation is real; greater anis because they do not live in family groups so indirect fitness is moot). 2. You will be able to clarify the misunderstanding that all social systems required a monogamy window – although never intended by Boomsma, this is how some people have misunderstood the phylogenies in Hughes et al. (2008). 3. That approach will put your own models in their right context for example when making clear that monogamy is not necessarily good if there is nest inheritance. 4. It will also allow connections with reproductive skew logic, which is about cooperative breeders that co-breed in spite of being totipotent (skew theory is not about permanent castes and obligate division of labor). 5. You might briefly refer to a discussion between Charlie Cornwallis & co, Dustin Rubinstein & co, and Michael Griesser & co on what drives cooperative breeding in birds. You will notice that what seemed a fundamental disagreement (relatedness first versus ecology first) now appears to be evaporating because Michael Griesser talks about extended parental care (no indirect fitness as in greater anis), and Cornwallis and Rubinstein recently published a joint paper agreeing on (as a broad trend) monogamy (low promiscuity) having promoted the colonization of harsh environments, but that living and radiating in harsh environments induced secondary reversals to promiscuity, some of that related to 'family life' giving way to groups of non-relatives. 6. In fact, all the six points towards the end of your paper are about cooperative breeding (none is about obligate worker castes) so all you need to do is make that explicit. 7. It would get you away from unfruitful discussions on whether inclusive fitness logic explains everything. Maternal manipulation or not, the key point is that helpers always obtain non-trivial indirect fitness benefits when they invest in helping (as they do in your refs 17-20; **no they do NOT always gain significant indirect fitness!**), which is all that matters when altruism is expressed in a facultative (context dependent) manner. It would seem to me that an emphasis in this direction would make your essay much more valuable because you could show that: 1. There is no fundamental disagreement between almost everybody that different types of models serve complementary purposes. 2. That different types of models only do that when previous insights are respected (provided they were thoroughly documented – NTW did not show that respect), and 3. That monogamy in the sense of low promiscuity is a statistical facilitator for cooperative breeding, not a necessary condition as it apparently was for evolving obligate 'neuter' castes or, as termite researchers would say, for evolving 'true workers'.

I would say that across the dueling papers, the proper conclusion is yes, monogamy can act as a facilitator under some sets of condition. However,

this is a far cry from the “universality” claim made in comments above. Both exercises in modeling and field data require a more careful consideration of the role high relatedness might play.

- I find much of this so called ‘controversy’ frustrating, because I do not feel responsible for it although I am happy to take credit for the monogamy hypothesis. If the author would care to re-read the closing paragraph of Boomsma (2009) I hope he can see why. I would now not have phrased this closing statement in sociobiology ‘eusociality language’, but by contrasting Wheeler-superorganismality (obligate unmated altruism) against cooperative breeding (among totipotent reproductive agents expressing at best facultative altruism) as done in later reviews. However, the meaning remains the same. The cooperative breeder domain of social evolution is more taxonomically diverse, more dynamic, and more variable in causation. Both models and empirical evidence show that and it is by acknowledging that distinction that a more transparent version of your essay can make a positive difference.

I don't think Boomsma is responsible for the controversy except in the sense that he put out a specific prediction/conclusion that others have challenged with mathematical representations. But, as I have repeatedly stated, I fundamentally differ from Boomsma in that I view facultative and obligate sterility as a distinction without an evolutionary difference.

The majority of Boomsma's general recommendations are to write a paper that agrees and reflects his particular world view. I, however, do not share large portions of that view. He has expressed his views, at great length, in several published papers. What I am asking for is to represent an alternative opinion in rather a short comment.

Specific points:

Line 16. ‘arguably intensified’ - I think this phrasing needs adjustment. To my knowledge no new authors have joined the small group of inclusive fitness theory critics. So we would need citations here proving that statement.

If one looks at the authors cited publishing with Nowak or E. O. Wilson just in this comment, one sees new names joining in on subsequent papers. The fact that it was suggested that I write this comment in the first place stems from another instance of a back & forth between Olejarz et al and Davies & Gardener. I guess that I too can be added to the list of growing critics.

Line 17. ‘aggressively vociferous’ - This kind of language should be avoided. Overall, it would be sufficient if lines 15-17 merely stated that no consensus has been reached. Who started making this debate acrimonious is open to interpretation. As I explained above, most opinionated evolutionary biologists considered NTW (2010) to be an unduly aggressive and poorly argued

attempt to discredit the most successful general theory of adaptation produced in the 20th century.

Revised to be toned down. (Although I would point out that Boomsma's comments about NTW in this review could be fairly called aggressively vociferous! But they do tend to stay out of the peer-reviewed lit.)

Line 33. To keep the discussion general, I think this should say 'as from multiple mating or cooperative breeding of multiple females'

Changed.

Line 39. 'touted' is unnecessary pejorative. Another case where language moderation is in order. The statement also seems to be formally incorrect. The 'monogamy window' is a theoretical concept (Boomsma, 2007, 2009, 2013) that has only been claimed to be a necessary condition for major transitions to coloniality with permanent castes (see general comments above).

Websters dictionary: Tout – to praise or publicize lavishly. As a word, I don't see it as pejorative, but I did change it to "presented".

Line 42. 'above phylogenies'? – you mean phylogenetically controlled comparative analyses?

Yes, changed.

Line 45. 'for unexplained reasons'?? It has been known for many years (Boomsma & Ratnieks, 1996; Strassmann, 2001) that obligate multiple queen mating in ants and corbiculate bees is a secondary derived evolutionary development, later confirmed to also apply to the vespine wasps by the studies of Foster and Ratnieks, all well before Hughes et al. (2008). The monogamy hypothesis provides a simple parsimonious general explanation for these convergent phenomena by stipulating that once the point of no return to irreversibly unmated worker castes had been passed, these secondary developments were free to evolve when they provided lasting benefits. If workers no longer have the plasticity to opt out by mating and dispersing, then one should not expect any group disharmony to arise except as a secondary consequence of chimeric colonies (requiring for example worker policing as documented most thoroughly in the honey bee). This point has been made in a single discussion sentence by Hamilton (1964), somewhat more elaborately by Boomsma & Ratnieks (1996), and explicitly in all the monogamy review published since 2007. That not all later empirical papers have been precise on this point cannot be held against the hypothesis having been transparent. It is actually at this level that the authors 'heterosis' hypothesis works.

High relatedness is touted as a predisposing factor towards increasing cooperation. If so, why is not then lowering relatedness a predisposing factor

towards increasing conflict? This is my challenge to kin selection advocates as there always seem to be rationalizations why lowering relatedness does not produce conflict. Remember, I am not convinced with the “irreversible” contention that some species are trapped into reproductive sterility.

Line 47-55. Terms like ‘cast doubt’ and ‘very maladaptive’ are inconsistent with the evidence and would thus best be phrased more carefully. This section claims there is repeated evidence for worker behaviour that does not make sense from an inclusive fitness perspective. I have checked all four studies and cannot see how they provide strong evidence for the inference that Hamiltonian logic is problematic. Nonacs et al. (2006) tested reproductive skew models that the author later dismissed as having limited general value (Nonacs & Hager, 2011; Biological Reviews). Skew models are derived from inclusive fitness logic but failed to live up to expectations because they needed too many special assumptions that could hardly be tested.

Not true. The most basic assumption and therefore the strongest prediction from skew theory was that relatedness was of paramount importance to predicting group interactions. This was almost never found to be the case. All the models to date have been based on Hamiltonian first principles and modeled in explicitly inclusive fitness terms; not special assumptions. The models’ failure must inevitably reflect back on how they were constructed and their claim to generality.

It is thus no great surprise that the qualitative cross-taxa predictions in the 2006 study appeared to work, but the detailed population-level predictions did not. The second study (Rehan et al., 2014) is about a bee where two sisters may build joint nests and where indirect fitness of subordinates was only half the direct fitness of nesting alone. However, that study did not measure the mortality costs of dispersing and initiating a nest alone. If that cost would be twice the failure rate of staying as a helper, the two options would be equivalent and consistent with Hamilton’s rule (as a mixed ESS not unlike the birds in your ref 14). The Gadagkar (2016) paper is essentially a review of decades of work and ends with the conclusion: ‘Clearly, Hamilton’s rule provides a powerful framework for understanding the evolution of social and altruistic behaviour in *R. marginata*. Even though I have often complained that most investigators focus excessively or exclusively on *r* and neglect the *b* and *c* terms in Hamilton’s rule, the power of the theory is evident when all three factors are considered, as I have attempted to do for *R. marginata*.’

And the title of the paper is: “Evolution of social behaviour in the primitively eusocial wasp *Ropalidia marginata*: do we need to look beyond kin selection?”, with the answer being yes.

Kapheim et al. (2015) is the best and most recent single case study, but also here authors had to infer what direct fitness helpers could have had in a rather indirect way. This allowed them to quantify productivity alternatives, but they could not consider the possibility that dispersal rather than helping would likely incur significantly higher mortality costs. In sum, none of these studies refute inclusive fitness logic because helpers obtained measurable inclusive fitness benefits in all cases. Such qualitative confirmations are often the best that can be achieved, because quantifying the Hamiltonian b and c variables remains essentially impossible in natural settings where dispersal costs are almost impossible to obtain. I suggest the author adjusts the phrasing of this paragraph to do justice to what was really shown. As far as I can see not all authors of these four papers have invoked maternal manipulation and Kapheim et al. only inferred that mechanism was consistent with the data. It has not been proven.

Kapheim et al (I am a coauthor here) had survival data for 180 nests in the field from the point where a nest was initiated to where it finally failed. The only survival data missing was the brief period of time where a female leaves her natal nest to point where she initiates her nest. This period may be as short as a few days. Based on productivity data from the 180 nests, we could calculate what the dispersal mortality rate would need to be for working and dispersing to have equal fitness. In essence, that rate had to be several orders of magnitude greater than what was observed across any part of the nesting period. Such a mortality rate would mean the entire *Megalopta* population was dying at unsustainable rates and ought to have gone to extinction in 2 generations.

Line 56. Having reached this point it is really hard to see the 'dissonance'. In all cases where tests were done (your refs 13, 14, 17-20) non-trivial indirect fitness benefits were always found, consistent with Hamiltonian logic. If there were quantitative mismatches (the indirect fitness benefits seeming not to be substantial enough), the most parsimonious explanation was that the non-measured costs of dispersal were substantial.

As argued above, the most parsimonious explanation is actually workers are manipulated into being workers.

Line 68-70. Yes of course, but this is the key argument of the monogamy window hypothesis for the emergence of life-time unmated castes. Under strict monogamy the r -terms cancel out of Hamilton's rule, so all that matters is the $b > c$ condition – i.e. what you call life-history and ecology. Before lineages reach that point of no-return what matters if $br > c$. This may seem a subtle difference, but it is not, because many combinations of b and r can make the inequality work. You just need substantial benefits applying in particular nest settings (and not in others) to express conditional altruism even when relatedness is modest. But altruism will never become obligate in

such scenarios. That apparently only happened when relatedness to siblings remained identical to relatedness to offspring over a huge number of generations, i.e. a sufficient number to irreversibly rewire complete developmental pathways (with hideously complex gene regulation networks) to produce permanent queen and worker castes. The assumption of a worker taking over the nest from her dead mother has never been part of the predictive domain of the monogamy window hypothesis. It is clearly a derived trait in lineages where permanent castes evolved, and as far as nest inheritance happens in societies with facultative altruism it may well prevent they can reach the monogamy window needed to evolve permanent castes. I believe that is consistent with your models, but phrasing this precisely matters for transparency of argument.

In response to the previous round of criticism, this paragraph deals explicitly with how the origin of cooperative behavior (not castes) arose and could be affected by monogamy – and why the models that dealt with that specific evolutionary event came to different predictions.

Line 75-76. The point here is whether maternal manipulation can induce irreversible transitions to life-time true-worker castes. There is no evidence to my knowledge that this has been modelled or observed to happen. Also the halictid systems considered in the Quinones & Pen model ultimately have to converge on strict monogamy before true helper traits can go to fixation (i.e. show the first signs of becoming irreversible).

This is a frustrating comment. My paragraph describes the results of sets of simulation models from several papers that explicitly examine the issue of the reproductive fate of developing offspring. One paper (mine) found that it easily occurs with maternal manipulation. What is wrong with that math?!

Line 78. 'also considered'? unclear phrasing. You mean 'considered a scenario'?

Yes – changed.

Line 82-87. I think this summary of the Davies and Gardner paper is understating the clarity of their results to a degree that does not serve the balanced view the author says he pursues. The first sentence will never communicate to the unprepared reader that their analyses showed that none of the conclusions of the Olejarz model really held up when realistic population genetics were used. The second sentence is totally non-transparent and can easily be read as the Davies and Gardner paper having unresolved loose ends, which I do not think was the case within the framework they considered. I also remember the Davies and Gardner paper showing the argument of worker sons not to hold up in their analyses, so more clear language (and references) would be needed if you disagree.

As Davies is the other reviewer on this comment and seems to be comfortable with my description, I believe I will leave it as is.

Line 89. Once more – the original monogamy hypothesis was primarily about the evolution of life-time unmated castes in the ants, bees, wasps and termites. Only after making that first-principle inference did Boomsma (2007) suggest that similar tendencies might be occurring in cooperative breeders, i.e. make the expression of facultative altruism more likely. It was that latter part that was tested and found to be correct in cooperatively breeding birds and mammals (Cornwallis; Lukas). Your phrasing is so general that this key point is glossed over, which muddles the water. In cooperative breeders monogamy is a statistical predictor; for the evolution of true workers it becomes a necessary condition. These are very different things (see general comments above).

Once more, this paper is NOT just about the 2007 version of the monogamy hypothesis!

Line 90-91. ‘enhance, retard, or have no effect ...’ as ‘overall conclusion’. Of what, of models that did not make their assumptions explicit? I am not aware of any empirical data set that has shown that the monogamy hypothesis was incorrect if you consider (as you should) that it never intended to replace Hamilton’s rule in vertebrate cooperative breeders and social insects where altruism is facultative (condition dependent).

Looking across all the times monogamy has been modeled, there is the variety of results. In no case was the actual math done in those papers shown to be flawed.

Line 92-95. This is true, but it seems to be just restating Hamilton’s rule. No monogamy paper ever claimed to replace that logic.

I repeat the above comment.

Line 96-105. My compliments for the constructive language in this paragraph. This is what the tone should be throughout this essay. You might want to add some of the 2010-2011 papers by Van Veelen as good examples of acrimonious inclusive fitness critique.

Line 106-113. It is unclear to me why the tone of this paragraph then becomes unbalanced again. I think ‘prosecutorial’ is once more a type of language to be avoided. All these papers did was reason from first GENERAL principles against a paper that was broadly conceived as a very unreasonable attack on massive progress in general understanding of social evolution based on inclusive fitness theory (see my general intro comments). It was NTW who started ignoring all the evidence (which you acknowledge in the previous paragraph). Objections were not ‘disingenuous’ (again wording

that should be removed). All we can say is that there is disagreement about whether single-locus modelling with strong discontinuous effects on assumed phenotypes is useful or not (see my general comment above). Whether the mathematics was correct is besides the point. The equations were correct in all papers as far as I know, else criticism would have been much stronger. The point is that when assumptions are wrong, the outcome will make no sense. In such cases results are not inconvenient (another value laden term to be reconsidered), but irrelevant for what happens in natural populations. Please rephrase this paragraph to achieve balanced coverage as you do in the preceding paragraph.

The balance requested by Boomsma is quite odd. Apparently I am an excellent writer when I find flaws with NTW and similar papers, but become unbalanced when I point out anything problematical about inclusive fitness and models and the papers written to defend it.

I will not change the word “disingenuous” unless it is to say “lied” or “dissembled”. The papers I cite never engaged substantively with my models or findings. Instead, they cavalierly dismissed the results for reasons that were either entirely irrelevant to what I did or outright attributed something to the models that was not a part of them. Boomsma is right that Nowak, Wilson and their immediate colleagues write papers that seem geared to inflame rather than generate a positive discussion. But I see the same utterly dismissive tone in many of the responses to those papers that are equally discouraging to reasoned discussion. As someone who has seen his work grossly misrepresented, I feel justified in calling out the papers that do so.

Please note that there is a critical distinction between disagreement and disparagement. This review by Boomsma show lengthy disagreement with what I have written. But in all instances Boomsma criticizes accurately what I say. He is fair (even if I think he is also wrong). In this comment and revisions, I make no attempt to hide my disagreements either, but always in response to what authors actually said or claimed. When in review, something I wrote was thought to be misleading, I revise to not be so. From personal discussions with colleagues on both sides of this question, I know there is anger that the ‘other’ side misrepresents their work just to score rhetorical points.

Line 112. ‘... exceptions accumulate’ I am not aware of such accumulation. Do you mean the same four papers addressed above (refs 17-20)? You would have to give new references to justify ‘accumulate’.

Models and data. Let’s revisit this in a decade from now! I suspect there are more studies out there that will accumulate the evidence.

Line 113. ‘uncritical’? To my knowledge almost every test of inclusive fitness theory has been an independent challenge of the general applicability of first-principle inclusive fitness theory. Critical attitudes in the Popperian sense

have thus abounded and the theory has survived these refutation assaults at least in a qualitative sense. You may find that having more quantitative matches would be desirable, and I would agree. However, even the qualitative matches have been inspiring enough to generate new avenues of derived inclusive fitness theory, as for example David Haig's genomic imprinting theory. By the way, also Dave Queller and Steve Frank have always argued that both types of models are useful and complementary. Changed uncritical to "immediate".

Line 114. Remove the value-laden word like 'presciently'

This is a word in praise of Okasha, who I think was prescient and the first to try to move this from an either/or argument into an "and" discussion. If more had taken his recommendations, I likely would not be writing this comment at all!

Line. 118-121. NTW did not pioneer simulation modelling and simulation modelling is something else than formal mathematical modelling. Formal mathematics provides first principle theory such as Hamilton's rule and its broadened version using the Price equation. Simulation models address specific cases where multiple restrictive assumptions need to be made, and where complexity exceeds what can be tracked analytically. As I tried to explain in my general comments above, simulation approaches can be useful but only if the assumptions about what phenotypes do are reasonable, and they should never claim to demonstrate something truly general. Overselling in that way was typical for the NTW paper and because their assumptions were often wrong or irrelevant, it gained no followers. I do not understand why the author is so adamant to defend this paper as some kind of a 'flagship case'. It detracts from the merits of his own models, which are more realistic and where he is willing to be specific about assumptions and about matches with general theory.

I appreciate (and agree!) that my papers are better attempts at trying to understand a phenomenon rather than to bash an entire field of study. Although I feel NTW is still the flagship work for arguing why one needs to model cooperation using a different methodology I am fine with not giving them credit for it. I deleted that clause.

Line 124-125. 'could consider'. Yes, but again the power issue is fundamentally different for lineages where colonies have facultative altruism versus those that have obligate altruism. You should make explicit that your point is about the former, not the latter type of social systems, and cite the well-known review by Beekman and Ratnieks to build any new argument on that synthesis. In fact it turns out that not only this first point, but also the following points 2-6 are about social systems with facultative altruism (see my last general comment above).

I did not mean to imply that I am the first person to ever think of any of the issues I raise. It is that they need more attention in my opinion. The B&R paper is a good one to cite, which I do now.

Line 136. 'can easily select' As far as I am aware there is no evidence for this to be 'easy' and to my knowledge there are no empirical data showing that selection for genetic diversity has produced consistent group adaptations in cooperative breeders. Reference 41 reports patterns across empirical studies but without addressing whether drifting 'workers' are potential reproducers or not. The only exceptions of unambiguous group adaptations owing to higher (chimeric) genetic diversity within groups appear to have happened after the point of no return to superorganismality had been passed. I.e., honeybee and leaf-cutting ant colonies deal better with disease when they have more patrines and costly obligate multiple queen mating is therefore a functional adaptation. Whether drifting workers can be considered to be a group adaptation remains to be proven and seems unlikely from a general theoretical perspective – these phenomena are merely effects of something else that could easily not have a functional adaptive significance (see Williams, 1966, for an important discussion about function and effect). Referring to 'indirect reciprocity' is unhelpful if you do not add a few lines of explanation of how that concept connects to what this essay is about.

This refers to a simulation model, which shows that drifting and lowered relatedness do not require unusual or extreme conditions in order to spread through a population. I have clarified this to state that is a theoretical prediction. However, it is supported by how often low-relatedness societies evolve and the taxonomic commonness of drifting. I also explained more what is meant by indirect reciprocity in this case.

Line 140-141. The monogamy hypothesis has always emphasized life-time (obligate) unmatedness, not obligate sterility, which requires diploidy to be an automatic consequence of unmatedness.

The argument is relevant independent of whether or not workers can produce male eggs or no eggs at all. I am questioning here the pronouncement that this feature is irreversible. To turn the tables, how does the reviewer know that there is no way to reverse sterility? Show me the experiment.

Line 144-146. This is a hand-waving statement. The monogamy hypothesis predicts (both in the early reviews and more explicitly in the later ones, which you should cite if you aim to be balanced) that such reversals should not have happened. That prediction has been consistent with all available evidence so far (see Boomsma & Gawne 2018 and Smith et al. 2018 for updates), so a statement like this is misleading unless you can cite empirical papers that have provided evidence to the contrary. A more productive approach would be to make sure you acknowledge that the 'monogamy

window' hypothesis for major transitions to caste societies is something else than facilitating monogamy in cooperative breeders (see my general comments) – and focus on the latter.

It is hand waving on both sides of this question. Which, of course, makes it attractive for direct investigation!

Line 147. The monogamy (window) hypothesis is about how facultative altruism can be replaced by obligate altruism, not about cooperation.

To me, that just is an added level of cooperation.

Line 148-150. Yes, it should work both ways, but only if relatedness is a true variable as in cooperatively breeding vertebrates, Polistes wasps, halictid bees etc. Once more, the core idea of the monogamy hypothesis is that strict full-sibling relatedness makes the relatedness terms cancel.

However, they no longer cancel once you have multiple parents!

That was apparently a necessary condition for major irreversible transitions to higher organismal complexity (both colonial superorganismality and obligate multicellular organismality), but not a sufficient one (the Leggett et al. paper – your ref 7 – also makes this point). Only long-term consistent $b > c$ benefits will forge major irreversible transitions after the necessary condition is fulfilled and that is a tall order. This entire point 3 is fine if you make explicit it is about cooperative breeders only (just like the other 5 points).

Obvious by now that I do not agree here!

In such 'societies' monogamy is never strict. References 4 and 8 are perfectly up front about that. Reference 3 was not, but you cannot hold that against the monogamy hypothesis. It is easy to plot the major irreversible transitions on the Hughes et al (2008) phylogeny (the base of the corbiculate bees, the base of the higher termites, the base of the vespine wasps). What you say here applies basal to those transitions, not beyond them – the later monogamy reviews have been very explicit about that, but that logic is also in the 2007 and 2009 versions. Irreversibility of transitions to coloniality where all individuals belong to a complementary caste for life (Wheeler superorganismality) is not just an assumption, it is consistent with all comparative data both for superorganismality and obligate multicellularity (cf Smith et al. Nature E&E 2018, Fisher et al. Current Biology 2013, Boomsma & Gawne, 2018).

What is and is not a multicellular organism is getting more nuanced every day and not all MET's required genetic relatedness (e.g., mitochondria and chloroplasts symbioses with other cells).

Line 151-156. Yes, Polistes colonies and human groups are societies, albeit very different ones because we have cumulative culture driving our social evolution. So again this is a 'society' argument, but it has no bearing on the

strict type of monogamy window hypothesized to represent a necessary condition for the emergence of obligate altruism in superorganisms (which no longer are societies of totipotent individuals). Once more, it would be useful if you somehow made these distinctions explicit when you intend this essay to be a way forward.

That distinction is obvious throughout the comment and need not be made in every other paragraph!

Line 158. 'sometimes'? Sibling competition within groups is generally and principally important as Hamilton and May (1977), Steve Frank (JEB 2012), Stu West & Ido Pen, and Leggett et al. (your ref 7) have amply documented. The group benefits you have in mind here are just the Hamiltonian $b > c$ values. Also your point 5 is about family groups before permanent castes evolve.

Yes, sometimes as opposed to never or always. The negative consequences of living together with close kin have not always received as much attention as the positive.

Line 162. And so is point 6. The Riehl work is very interesting, but not as a challenge of the monogamy hypothesis for altruism driven by indirect fitness benefits, because two key conditions do not apply: 1. 'helpers' gain no inclusive fitness – they just hang around to receive prolonged parental investment. 2. These greater anis are not family groups where altruism could have evolved.

Yes, but monogamy (or not) is quite important in determining how inbreeding effects could obviate Hamilton's rule predictions.

Line 168-172. The final quote is nice, but should in my opinion refer to the need to test both the assumptions and the predictions of any model used to better understand social evolution. It has no specific bearing on ecology and group-level benefits, which are fully covered by Hamilton's rule as the series of Steve Frank papers in JEB (2011-2013) amply document. Does it have bearings on genetics? Yes, quite possibly so, but remember Hamilton's rule is about genetics as well. If you want to make the genetics more explicit that can be useful, but it will always imply making restrictive assumptions that the Price equation (general inclusive fitness) approach does not need to make. If you keep adding special assumptions in order to gain 'realism', you may end up with a model that is very precise about a scenario that may never apply in nature. See my general comments and the Queller quote given above on how generality and precision of models trade-off. Both approaches are valuable when assumptions about individual and colony phenotypes are reasonable – and then they are complementary ways towards understanding. Models with inappropriate or unspecified assumptions are not useful. I think that should be

a major condition for making your 'verify' by modelling credible, but the final arbiter of course remains what the empirical data show.

The counter argument is that general does not equal universal. General rules are fine, and Hamilton's rule is one of the best, but there are clear instances where $rb - c > 0$ fails to adequately describe what you have in nature. It is important to know where and when the general is contravened by the specific.